# Quantitative glycoproteomics reveals cellular substrate selectivity of the ER protein quality control sensors UGGT1 and UGGT2

Benjamin M Adams[1,2], Nathan P Canniff[1,2], Kevin P Guay[1,2],
Ida Signe Bohse Larsen[3,4], Daniel N Hebert[1,2]*

[1]Department of Biochemistry and Molecular Biology, University of Massachusetts, Amherst, United States; [2]Program in Molecular and Cellular Biology, University of Massachusetts, Amherst, United States; [3]Department of Cellular and Molecular Medicine, University of Copenhagen, Copenhagen, Denmark; [4]Copenhagen Center for Glycomics, University of Copenhagen, Copenhagen, Denmark

**Abstract** UDP-glucose:glycoprotein glucosyltransferase (UGGT) 1 and 2 are central hubs in the chaperone network of the endoplasmic reticulum (ER), acting as gatekeepers to the early secretory pathway, yet little is known about their cellular clients. These two quality control sensors control lectin chaperone binding and glycoprotein egress from the ER. A quantitative glycoproteomics strategy was deployed to identify cellular substrates of the UGGTs at endogenous levels in CRISPR-edited HEK293 cells. The 71 UGGT substrates identified were mainly large multidomain and heavily glycosylated proteins when compared to the general N-glycoproteome. UGGT1 was the dominant glucosyltransferase with a preference toward large plasma membrane proteins whereas UGGT2 favored the modification of smaller, soluble lysosomal proteins. This study sheds light on differential specificities and roles of UGGT1 and UGGT2 and provides insight into the cellular reliance on the carbohydrate-dependent chaperone system to facilitate proper folding and maturation of the cellular N-glycoproteome.

*For correspondence:
dhebert@biochem.umass.edu

Competing interests: The authors declare that no competing interests exist.

## Introduction

Protein folding in the cell is an error-prone process and protein misfolding is the basis for a large number of disease states (*Hebert and Molinari, 2007*; *Hartl, 2017*). A significant fraction of the proteome in mammalian cells passes through the secretory pathway by first being targeted to the endoplasmic reticulum (ER) where folding occurs (*Uhlén et al., 2015*; *Itzhak et al., 2016*; *Adams et al., 2019a*). Molecular chaperones of the ER help to guide secretory pathway cargo along a productive folding pathway by directing the trajectory of the folding reaction, inhibiting non-productive side reactions such as aggregation or by retaining immature or misfolded proteins in the ER until they can properly fold or be targeted for degradation. Understanding how chaperone binding controls the maturation and flux of proteins through the secretory pathway is of important fundamental biological concern and will impact our knowledge of protein folding diseases and the development of potential therapeutics including the production of biologics that are frequently secretory proteins.

Proteins that traverse the secretory pathway are commonly modified with N-linked glycans as they enter the ER lumen (*Zielinska et al., 2010*). These carbohydrates serve a variety of roles including acting as quality control tags or attachment sites for the lectin ER chaperones calnexin and calreticulin (*Helenius and Aebi, 2004*; *Hebert et al., 2014*). N-glycosylation commences co-translationally in mammals and the first round of binding to calnexin and calreticulin is initiated

shortly thereafter by the rapid trimming of glucoses by glucosidases I and II to reach their monoglucosylated state (*Chen et al., 1995*; *Cherepanova et al., 2019*). Lectin chaperone binding is multifunctional as it has been shown to: (1) direct the folding trajectory of a protein by acting as a holdase that slows folding in a region-specific manner (*Daniels et al., 2003*; *Wang et al., 2008*); (2) act as an adapter or platform to recruit folding factors including oxidoreductases (ERp57 and ERp29) and a peptidyl-prolyl *cis trans* isomerase (CypB) to maturing nascent chains (*Kozlov and Gehring, 2020*); (3) diminish aggregation (*Hebert et al., 1996*); (4) retain immature, misfolded, or unassembled proteins in the ER (*Rajagopalan et al., 1994*); and (5) target aberrant proteins for degradation by ER-associated degradation (ERAD) and ER-phagy (*Molinari et al., 2003*; *Oda et al., 2003*; *Forrester et al., 2019*). For glycoproteins, the lectin chaperones appear to be the dominant chaperone system as once an N-glycan is added to a region on a protein, it has been shown to be rapidly passed from the ER Hsp70 chaperone BiP to the lectin chaperones, further underscoring their central role in controlling protein homeostasis in the secretory pathway (*Hammond and Helenius, 1994*).

N-glycan trimming to an unglucosylated glycoform by glucosidase II supports substrate release from the lectin chaperones. At this stage, if the protein folds properly, it is packaged into COPII vesicles for anterograde trafficking (*Barlowe and Helenius, 2016*). Alternatively, substrates that are evaluated to be non-native are directed for rebinding to the lectin chaperones by the protein folding sensor UDP-glucose:glycoprotein glucosyltransferase 1 (UGGT1) that reglucosylates immature or misfolded proteins (*Helenius, 1994*; *Sousa and Parodi, 1995*). Since UGGT1 directs the actions of this versatile lectin chaperone system and thereby controls protein trafficking through the ER, it acts as a key gatekeeper of the early secretory pathway. Therefore, it is vital to understand the activity of UGGT1 and the scope of substrates it modifies.

Our current knowledge of the activity of UGGT1 relies largely on studies using purified components. UGGT1 was found to recognize non-native or near-native glycoproteins with exposed hydrophobic regions using in vitro approaches where the modification of glycopeptides, engineered or model substrates by purified UGGT1 was monitored (*Ritter and Helenius, 2000*; *Taylor et al., 2003*; *Caramelo et al., 2004*). Recent crystal structures of fungal UGGT1 have shown that it possesses a central, hydrophobic cavity in its protein sensing domain, which may support hydrophobic-based interactions for substrate selection (*Roversi et al., 2017*; *Satoh et al., 2017*).

Cell-based studies of UGGT1 have relied on the overexpression of cellular and viral proteins (*Soldà et al., 2007*; *Pearse et al., 2008*; *Ferris et al., 2013*; *Tannous et al., 2015*). *Uggt1* knockout studies have found that the roles of UGGT1 appear to be substrate specific as UGGT1 can promote, decrease, or not affect the interaction between substrates and calnexin (*Soldà et al., 2007*). Prosaposin, the only known cellular substrate of UGGT1 when expressed at endogenous levels, grossly misfolds in the absence of *Uggt1* and accumulates in aggresome-like structures (*Pearse et al., 2010*). Work in animals has further emphasized the importance of UGGT1 as the deletion of *Uggt1* in mice is embryonically lethal (*Molinari et al., 2005*).

UGGT1 has a paralogue, UGGT2, but it has not demonstrated cellular activity (*Arnold et al., 2000*). Domain swapping experiments have demonstrated that UGGT2 possesses a catalytically active glucosyltransferase domain when appended to the folding sensor domain of UGGT1 (*Arnold and Kaufman, 2003*). In vitro experiments using purified, chemically glycosylated interleukin-8 (IL-8), which is not glycosylated in cells, have found that UGGT2 can glucosylate IL-8 (*Takeda et al., 2014*). This suggests that UGGT2 may be an additional reglucosylation enzyme or protein folding sensor of the ER.

Unlike the classical ATP-dependent chaperones that directly query the conformation of their substrates (*Balchin et al., 2016*), binding to the lectin chaperones is dictated by enzymes that covalently modify the substrate (*Helenius and Aebi, 2004*; *Hebert et al., 2014*). Rebinding to the carbohydrate-dependent chaperones is initiated by the UGGTs that interrogate the integrity of the structure of the protein. Therefore, the proteome-wide detection of cellular UGGT substrates provides the unprecedented opportunity to identify clients that require multiple rounds of chaperone binding and are more reliant on lectin chaperone binding for proper maturation and sorting. Therefore, we designed a cell-based quantitative glycoproteomics approach to identify high-confidence endogenous substrates of UGGT1 and UGGT2 by the affinity purification of monoglucosylated substrates in CRISPR/Cas9-edited cells. UGGT1 and UGGT2 substrates were found to display multiple features of complex proteins including extended lengths plus large numbers of Cys residues and N-glycans.

Specific substrates of either UGGT1 or UGGT2 were also discovered, therefore determining that UGGT2 possessed glucosyltransferase activity and identifying its first natural substrates. UGGT1 demonstrated a slight preference for transmembrane proteins, especially those targeted to the plasma membrane, while UGGT2 modification favored soluble lysosomal proteins. The identification of reglucosylated substrates improves our understanding of their folding and maturation pathways and has implications regarding how folding trajectories may be altered in disease states.

## Results

### Experimental design

To identify the substrates that are most dependent upon persistent calnexin/calreticulin binding, we isolated and identified endogenous substrates of the ER protein folding sensors UGGT1 and UGGT2. As the product of a reglucosylation by the UGGTs is a monoglucosylated N-glycan, the presence of the monoglucosylated glycoform was used as a readout for substrate reglucosylation. N-glycans are originally transferred to nascent glycoproteins containing three glucoses, therefore a monoglucosylated glycan can be generated either through trimming of two glucoses from the nascent N-linked glycan or through reglucosylation by the UGGTs. In order to isolate the reglucosylation step from the trimming process, a gene edited cell line was created that transfers abbreviated unglucosylated N-linked glycans to nascent chains. The N-linked glycosylation pathway in mammalian cells is initiated through the sequential addition of monosaccharides, mediated by the *ALG* (Asn-linked glycosylation) gene products, to a cytosolically exposed dolichol-P-phosphate embedded in the ER membrane (*Aebi, 2013*; *Cherepanova et al., 2016*; *Figure 1A*). The immature dolichol-P-phosphate precursor is then flipped into the ER lumen and sequential carbohydrate addition is continued by additional ALG proteins. The completed N-glycan ($Glc_3Man_9GlcNAc_2$) is then appended to an acceptor Asn residue in the sequon Asn-Xxx-Ser/Thr/Cys (where Xxx is not a Pro) by the oligosaccharyl transferase (OST) complex (*Cherepanova et al., 2016*). Initially, a Chinese Hamster Ovary (CHO) cell line with a defect in *Alg6* was employed to establish the utility of this approach to follow (re)glucosylation (*Quellhorst et al., 1999*; *Cacan et al., 2001*; *Pearse et al., 2008*; *Pearse et al., 2010*; *Tannous et al., 2015*). As the CHO proteome is poorly curated compared to the human proteome, CRISPR/Cas9 was used to knock out the *ALG6* gene in HEK293-EBNA1-6E cells to provide a cellular system that transferred non-glucosylated glycans ($Man_9GlcNAc_2$) to substrates. In these *ALG6$^{-/-}$* cells, a monoglucosylated glycan is solely created by the glucosylation by the UGGTs providing a suitable system to follow the glucosylation process (*Figure 1B*).

To aid in substrate identification, an inhibitor of glucosidases I and II, deoxynojirimycin (DNJ), was added 1 hr prior to cell lysis to block glucose trimming and trap monoglucosylated products. Monoglucosylated substrates were then isolated by affinity purification using recombinant glutathione S-transferase-calreticulin (GST-CRT), as calreticulin binds monoglucosylated proteins. To account for nonspecific binding, a lectin-deficient construct (GST-CRT-Y109A) was used as an affinity purification control (*Kapoor et al., 2004*). Affinity purified substrates were reduced, alkylated, and trypsin digested. The resulting peptides were labeled with tandem mass tags (TMTs) (*Rauniyar and Yates, 2014*), deglycosylated using PNGaseF, and analyzed by mass spectrometry to identify substrates of the UGGTs. The use of TMT, as well as the control GST-CRT-Y109A affinity purification, allows for robust, quantitative identification of substrates of the UGGTs. The resulting data was analyzed by calculating the fold change in abundance of the TMT associated with proteins identified through affinity purification using wild-type (WT) GST-CRT over affinity purification using GST-CRT-Y109A. To be considered a UGGT substrate, a cutoff of threefold (WT GST-CRT/GST-CRT-Y109A) was applied. This conservative cutoff was set to give a high level of confidence in the identified substrates, as below this cutoff, increasing fractions of non-secretory pathway proteins were found.

### Substrate identification of the UGGTs

In order to determine the cellular substrates of the UGGTs, the above glycoproteomics protocol was followed using *ALG6$^{-/-}$* cells. A restricted pool of 37 N-linked glycosylated proteins was identified as substrates of the UGGTs (*Figure 1C* and *Supplementary file 1*). Prosaposin, the only previously known endogenous substrate of the UGGTs, was included in this group, supporting the utility of the approach (*Pearse et al., 2010*). Integrin β−1 showed the most significant fold change (WT GST-

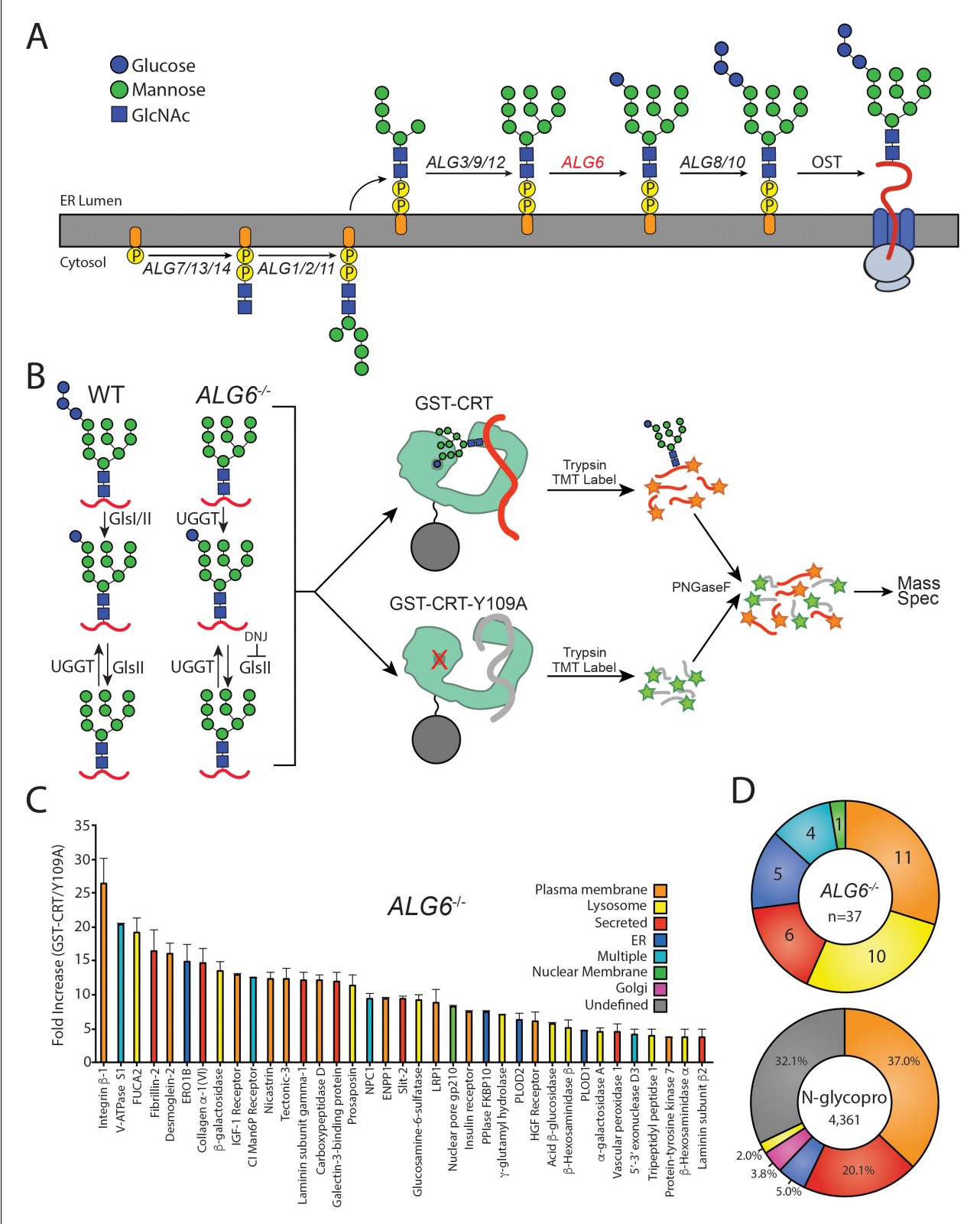

**Figure 1.** The identification of UDP-glucose:glycoprotein glucosyltransferase (UGGT) 1/2 substrates. (**A**) The pathway of N-glycosylation in eukaryotic cells is depicted. N-glycan synthesis is initiated in the outer endoplasmic reticulum (ER) membrane leaflet on a dolichol-P-phosphate facing the cytoplasm. Flipping of the precursor N-glycan to the ER luminal leaflet and further synthesis steps mediated by ALG proteins leads to eventual transfer of a $Glc_3Man_9GlcNAc_2$ N-glycan to a substrate by the oligosaccharyl transferase complex. ALG6 (red lettering) catalyzes the transfer of the initial

*Figure 1 continued on next page*

*Figure 1 continued*

glucose onto the $Man_9$ precursor N-glycan. (**B**) In wild-type (WT) cells, a $Glc_3Man_9GlcNAc_2$ N-glycan is transferred to substrates. Monoglucosylated substrates may therefore occur via trimming by glucosidases I/II (GlsI/II) or reglucosylation by UGGT1/2. In $ALG6^{-/-}$ cells, a $Man_9GlcNAc_2$ N-glycan is transferred to substrates. Therefore, monoglucosylated substrates may only occur through reglucosylation by UGGT1/2. Deoxynojirimycin (500 µM) was added to block the trimming of monoglucosylated substrates by GlsII. $ALG6^{-/-}$ cells were then lysed and split equally between affinity purifications with either GST-CRT or GST-CRT-Y109A bound to glutathione beads. Affinity-purified samples were then reduced, alkylated, trypsinized, and labeled with tandem mass tag (TMT) labels. Samples were then deglycosylated with PNGaseF, pooled, and analyzed by mass spectrometry. (**C**) Substrates were identified by dividing the quantification of the TMT label in the GST-CRT condition for each protein by that of the associated GST-CRT-Y109A condition, yielding the fold increase. Localization as predicted by UniprotKB annotation is depicted. A cutoff of threefold increase was applied. Data is representative of two independent experiments. Error bars represent standard error of the mean (SEM). (**D**) The N-glycoproteome (N-glycopro) was computationally determined by collecting all proteins annotated to contain N-glycans by UniprotKB. Annotated localization information was then used to computationally determine the localization distribution of the N-glycoproteome as well as the identified UGGT substrates.

The online version of this article includes the following source data for figure 1:

**Source data 1.** TMT quantification results for *Figure 1C*.

CRT/GST-CRT-Y109A) of ~26-fold, indicating there is a large dynamic range of reglucosylation levels.

The cell localizations of UGGT substrates were then determined by using their UniprotKB classification. Approximately two-thirds of the UGGT substrates are destined for the plasma membrane or lysosomes (*Figure 1C and D*). Additional substrates are secreted or are resident to the ER or nuclear membrane. Nuclear pore membrane glycoprotein 210 (NUP210) was the only nuclear membrane protein found to be reglucosylated and it is the sole subunit of the nuclear pore that is N-glycosylated (*Beck and Hurt, 2017*). The nucleus and ER share a contiguous membrane. Proteins targeted to the nuclear membrane are first inserted into the ER membrane, then move laterally to the nuclear membrane (*Katta et al., 2014*). Four proteins were designated as 'multiple localizations' including cation-independent mannose-6-phosphate receptor (CI-M6PR), which traffics between the Golgi, lysosome, and plasma membrane (*Dell'Angelica and Payne, 2001*).

To distinguish the general pool of substrates that the UGGTs are expected to be exposed to, N-glycosylated proteins of the secretory pathway proteome (N-glycoproteome) were computationally defined (*Supplementary file 2*). The N-glycoproteome is comprised of proteins that are targeted to the ER either for residency in the secretory/endocytic pathways or for trafficking to the plasma membrane or for secretion. The reviewed UniprotKB *H. sapiens* proteome (20,353 total proteins) was queried to identify all proteins annotated as N-glycosylated, resulting in a set of 4520 proteins. This set was then curated to remove proteins predicted to be mitochondrial, contain less than 50 amino acids or redundant isoforms. The resulting N-glycoproteome contained 4361 proteins, predicting ~21% of the proteome is N-glycosylated. Comparing UGGT substrates to the N-glycoproteome allows for the characterization of feature preferences of substrates for the UGGTs.

The majority of the N-glycoproteome was either localized to the plasma membrane (37%) or secreted (20%) according to their UniprotKB designations. Smaller fractions of the N-glycoproteome reside in the ER (5%), Golgi (4%), or lysosomes (2%). UGGT substrates are therefore significantly enriched for lysosomal proteins compared to the N-glycoproteome, while all other localizations display a similar distribution to their availability. In total, these results demonstrate the ability to identify substrates of the UGGTs proteomically and suggest that the UGGTs display substrate preferences.

## Determination of UGGT1- and UGGT2-specific substrates

There are two ER glucosyltransferase paralogues, UGGT1 and UGGT2, though currently there is no evidence that UGGT2 acts as a protein sensor or a glucosyltransferase in the cell. Therefore, we sought to determine if UGGT2 has glucosyltransferase activity in the cell, and if so, do these two paralogues have different substrate specificities. To address this concern, GST-CRT affinity purification and TMT mass spectrometry were used to identify substrates of UGGT1 in $ALG6/UGGT2^{-/-}$ cells and potential UGGT2 substrates in $ALG6/UGGT1^{-/-}$ cells.

With the $ALG6/UGGT2^{-/-}$ cells, 66 N-glycosylated proteins were identified as reglucosylation substrates using the three-fold cutoff (GST-CRT/CST-CRT-Y109A) (*Figure 2A*). Nearly double the number of UGGT1 substrates were identified through this approach compared to using $ALG6^{-/-}$ cells where both UGGT1 and UGGT2 were present. This expansion in substrate number is likely due to

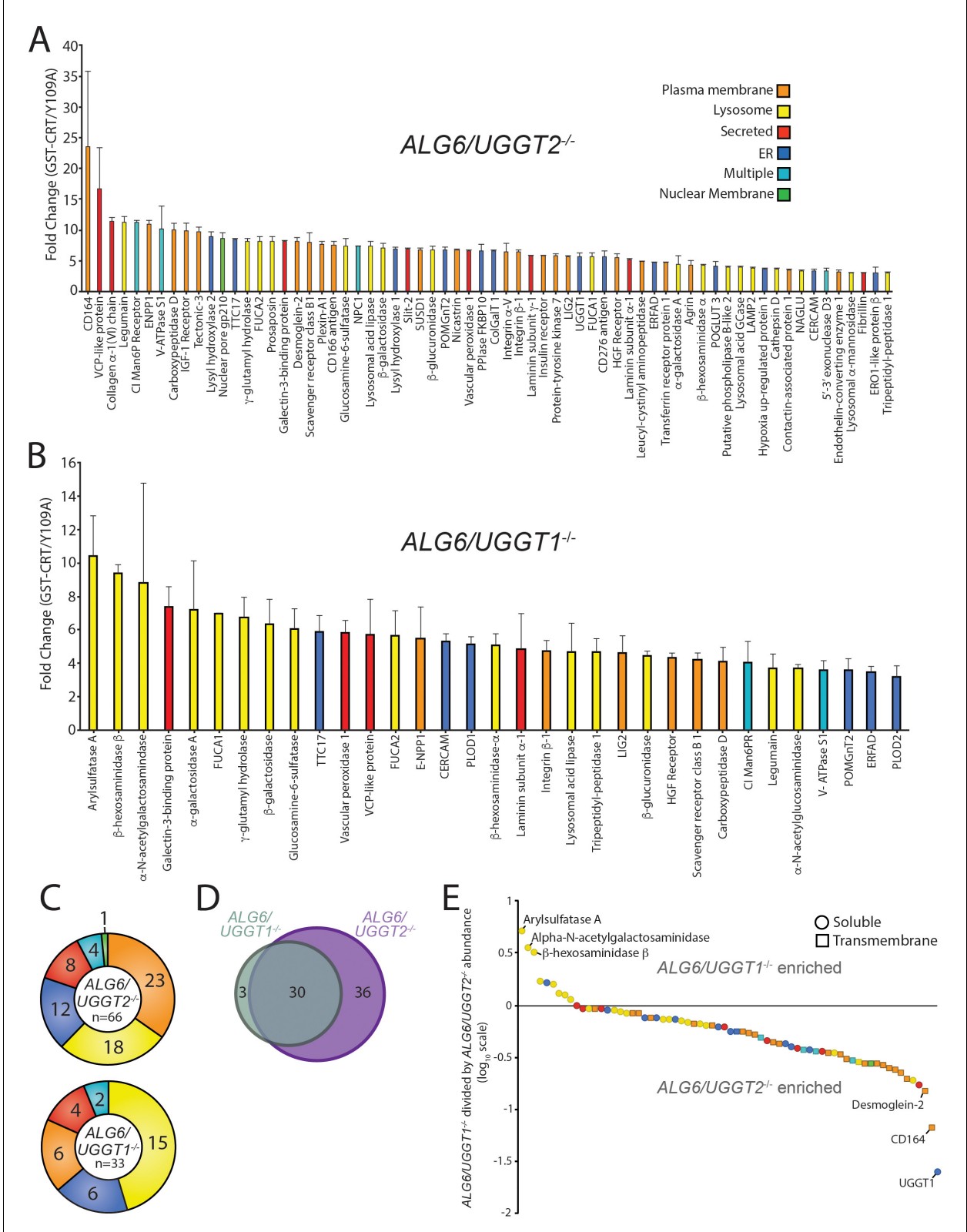

**Figure 2.** Identification of UDP-glucose:glycoprotein glucosyltransferase (UGGT)1- and UGGT2-specific substrates. (**A**) Reglucosylation substrates in *ALG6/UGGT2⁻/⁻* cells were identified and quantified as previously described in *Figure 1*. Localizations as annotated by UniprotKB are depicted. Data are representative of two independent experiments. Error bars represent SEM. (**B**) Reglucosylated substrates in *ALG6/UGGT1⁻/⁻* cells were identified and quantified as previously above. (**C**) The distribution of localizations as annotated by UniprotKB for reglucosylation substrates identified in both

*Figure 2 continued on next page*

Figure 2 continued

*ALG6/UGGT2⁻/⁻* and *ALG6/UGGT1⁻/⁻* cells is depicted. (D) The overlap of reglucosylated substrates identified in both *ALG6/UGGT2⁻/⁻* cells (purple) and *ALG6/UGGT1⁻/⁻* cells (gray) is visualized by a Venn diagram. (E) Reglucosylated substrate enrichment in either *ALG6/UGGT1⁻/⁻* or *ALG6/UGGT2⁻/⁻* cells is depicted by dividing the tandem mass tag quantification for each protein in *ALG6/UGGT1⁻/⁻* cells by the associated value in *ALG6/UGGT2⁻/⁻* cells on a $\log_{10}$ scale. Positive and negative values represent enrichment in *ALG6/UGGT1⁻/⁻* and *ALG6/UGGT2⁻/⁻* cells, respectively. Localization (coloring) and topology (soluble [circles] or transmembrane [squares]) are depicted based on UniprotKB annotation.

The online version of this article includes the following source data and figure supplement(s) for figure 2:

**Source data 1.** TMT quantification results for *Figure 2A* and *Figure 2B*.
**Figure supplement 1.** UDP-glucose:glycoprotein glucosyltransferase (UGGT)1 and UGGT2 expression.
**Figure supplement 2.** mRNA expression analysis of UDP-glucose:glycoprotein glucosyltransferase (UGGT)1 and UGGT2 substrates.
**Figure supplement 3.** β-hexosaminidase subunit β trafficking and hypoglycosylation and CI-M6PR hypoglycosylation.

the ~50% increase in expression of UGGT1 in *ALG6/UGGT2⁻/⁻* cells (*Figure 2—figure supplement 1*). The substrate demonstrating the most significant fold change (23.5-fold) was CD164, creating a similar dynamic range for reglucosylation to that observed in *ALG6⁻/⁻* cells.

To identify possible UGGT2-specific substrates, *ALG6/UGGT1⁻/⁻* cells were used to isolate UGGT2 modified substrates. Thirty-four proteins passed the threefold GST-CRT/GST-CRT-Y109A cutoff, with 33 of these proteins predicted to be N-glycosylated and localized to the secretory pathway (*Figure 2B*). Importantly, this demonstrated for the first time that UGGT2 was a functional glycosyl-transferase capable of reglucosylating a range of cellular substrates. The glycoprotein with the most significant fold change was arylsulfatase A (10.4-fold). Notably, eight of the nine strongest UGGT2 substrates, or 15 of 33 substrates overall, are lysosomal proteins (*Figure 2B and C*). While UGGT1 was also observed to engage a significant percentage of lysosomal proteins (27%), 45% of UGGT2 substrates are lysosomal. Both of these percentages are significantly enriched when compared to the N-glycoproteome for which only 2% is comprised of resident lysosome proteins (*Figure 1D*).

UGGT1 substrates were enriched for plasma membrane localized proteins (35%) when compared to UGGT2 substrates (18%), while plasma membrane proteins were found to compose a similar per-cent of the N-glycoproteome (37%) compared to UGGT1 substrates. Similar percentages of UGGT1 and UGGT2 substrates localize to the ER (18%), are secreted (12%), or are found in multiple localiza-tions (6%) (*Figure 2C*). Even though 4% of the N-glycoproteome is composed of Golgi proteins (*Figure 1D*), neither UGGT1 nor UGGT2 appeared to modify Golgi localized proteins.

The number of UGGT1 substrates was double that of UGGT2 suggesting that UGGT1 carried the main quality control load. Only 3 out of 33 UGGT2 substrates were specific to UGGT2. These three UGGT2-specific substrates included arylsulfatase A, α-N-acetylgalactosaminidase, and β-hexosamini-dase subunit β (HexB), three soluble lysosomal enzymes (*Figure 2D and E*). Thirty substrates over-lapped between UGGT1 and UGGT2, while 36 substrates were found to be specific to UGGT1 (*Figure 2D* and *Supplementary file 3*). The preference for the shared substrates was explored by plotting all proteins identified as a substrate of either glucosyltransferase on a $\log_{10}$ scale of the associated TMT value in *ALG6/UGGT1⁻/⁻* cells divided by the values in *ALG6/UGGT2⁻/⁻* cells (*Figure 2E*). Proteins enriched as UGGT2 substrates therefore possess positive values while UGGT1 enriched substrates have negative values.

The three substrates found to be specific to UGGT2 clustered away from all other proteins (*Figure 2E* at the top left). The remaining UGGT2 enriched substrates, except for one ER localized protein, localized to the lysosome. All the UGGT2 favored substrates were soluble proteins. In con-trast, UGGT1 favored proteins were greater in number and displayed a diversity of localizations with a preference for plasma membrane proteins. These results indicate that UGGT2 is a functional gluco-syltransferase, which preferentially engages soluble lysosomal proteins while UGGT1 modifies a wider variety of proteins with a preference for plasma membrane and transmembrane domain-con-taining proteins in general.

## Validation of UGGT substrates

Having identified numerous novel substrates of the UGGTs, a select number of these substrates was tested for reglucosylation to validate the identification approach. Substrates were chosen based on a diversity of topologies, lengths, differences in propensities as UGGT1 or UGGT2 substrates, and reagent availability. Monoglucosylated substrates were affinity isolated from *ALG6⁻/⁻*, *ALG6/*

UGGT1$^{-/-}$, ALG6/UGGT2$^{-/-}$, and ALG6/UGGT1/UGGT2$^{-/-}$ cells using GST-CRT compared to GST-CRT-Y109A. Substrates were then identified by immunoblotting with the percent reglucosylation determined by subtracting the amount of protein bound by GST-CRT-Y109A from that of GST-CRT, divided by the total amount of substrate present in the whole cell lysate (WCL), and multiplying by 100.

CI-M6PR and insulin-like growth factor type one receptor (IGF-1R) are both large type I membrane proteins that possess multiple N-glycosylation sites (*Figure 3D and H*). Overall, 10% of CI-M6PR was reglucosylated in *ALG6$^{-/-}$* cells (*Figure 3B*). The modification level of CI-M6PR was significantly reduced in *ALG6/UGGT1$^{-/-}$*, but not *ALG6/UGGT2$^{-/-}$* cells. As a control, reglucosylation was not observed in *ALG6/UGGT1/UGGT2$^{-/-}$* cells. A similar profile was observed for IGF-1R where reglucosylation levels reached 12% in *ALG6/UGGT2$^{-/-}$* cells (*Figure 3E–G*). Altogether, these findings were consistent with the quantitative glycoproteomics isobaric labeling results (*Figure 3C and G*), confirming that CI-M6PR and IGF-1R are efficient substrates of UGGT1.

Next, the reglucosylation of the type II membrane protein, ectonucleotide pyrophosphatase/phosphodiesterase family member 1 (ENPP1) was analyzed (*Figure 3L*). ENPP1 was found to be reglucosylated at similar levels in *ALG6$^{-/-}$* (7%) and *ALG6/UGGT1$^{-/-}$* (7%) cells. In *ALG6/UGGT2$^{-/-}$* cells, reglucosylation increased to 12%, while in *ALG6/UGGT1/UGGT2$^{-/-}$* cells reglucosylation decreased to 1% (*Figure 3I and J*). These results suggest that ENPP1 can be reglucosylated by both UGGT1 and UGGT2, with a slight preference for UGGT1, supporting the TMT mass spectrometry results (*Figure 3K*).

The reglucosylation of the smaller soluble lysosomal protein, HexB, was also tested (*Figure 3M–P*). HexB is processed into three disulfide-bonded chains in the lysosome (*Mahuran et al., 1988*). Only immature or ER localized proHexB was affinity purified by GST-CRT (*Figure 3M*, lanes 2, 5, 8, and 11). HexB was reglucosylated at 34% in *ALG6$^{-/-}$* cells (*Figure 3N*). No significant change in glucosylation levels occurred when UGGT1 was also knocked out (35%). However, a reduction to 20% reglucosylation of HexB was observed in *ALG6/UGGT2$^{-/-}$* cells, and complete loss of reglucosylation was seen in *ALG6/UGGT1/UGGT2$^{-/-}$* cells. *ALG6/UGGT1$^{-/-}$* cells consistently displayed increased levels of expression of HexB (*Figure 3M*, lane 4), and this was consistent with RNAseq data (*Figure 2—figure supplement 2B*). These results confirm the mass spectrometry results that showed HexB to be a favored substrate of UGGT2 (*Figure 3O*). It is also notable that HexB, as the first validated substrate of UGGT2, is highly reglucosylated. As reglucosylation was not observed for any of the validated substrates tested when both UGGT1 and UGGT2 were knocked out, these glucosyltransferases appear to be responsible for the reglucosylation of N-glycans in the ER. Taken together, these results demonstrate that the mass spectrometry screen accurately identified substrates of the UGGTs, as well as differentiated between substrates specific to either UGGT1 or UGGT2.

## Analysis of UGGT substrates

To investigate the properties of the substrates modified by the UGGTs and identify potential types of proteins UGGT1 and UGGT2 modify, a systematic analysis of the substrates of the UGGTs was performed and compared to the general properties of the N-glycoproteome. All characteristics were analyzed using UniprotKB annotations. Initially, the length of substrates was compared to the N-glycoproteome. The N-glycoproteome ranged widely in size, from elabela (54 amino acids) to mucin-16 (14,507 amino acids). The overall amino acid distribution of the N-glycoproteome was significantly shifted smaller compared to the size of UGGT substrates (*Figure 4A*). The median size of the N-glycoproteome was 443 amino acids, compared to 737 for UGGT substrates found in *ALG6$^{-/-}$* cells. Substrates of both UGGT1 (718 amino acid median) and UGGT2 (585 amino acids) are significantly larger when compared to the N-glycoproteome. This increase in length may lead to more complex folding trajectories, requiring increased engagement with the lectin chaperones for efficient maturation.

The distribution of the number of N-glycans possessed by the N-glycoproteome (median of two glycans per glycoprotein) was also shifted significantly smaller than that of UGGT1 (seven glycans) or UGGT2 (five glycans) substrates (*Figure 4B*). All the UGGT substrates displayed a decreased density of proteins at low N-glycan content values that are heavily populated in the N-glycoproteome. Despite the identification of UGGT1 and UGGT2 substrates generally containing high numbers of N-glycans, multiple substrates possessed as few as two N-glycans, suggesting that the experimental

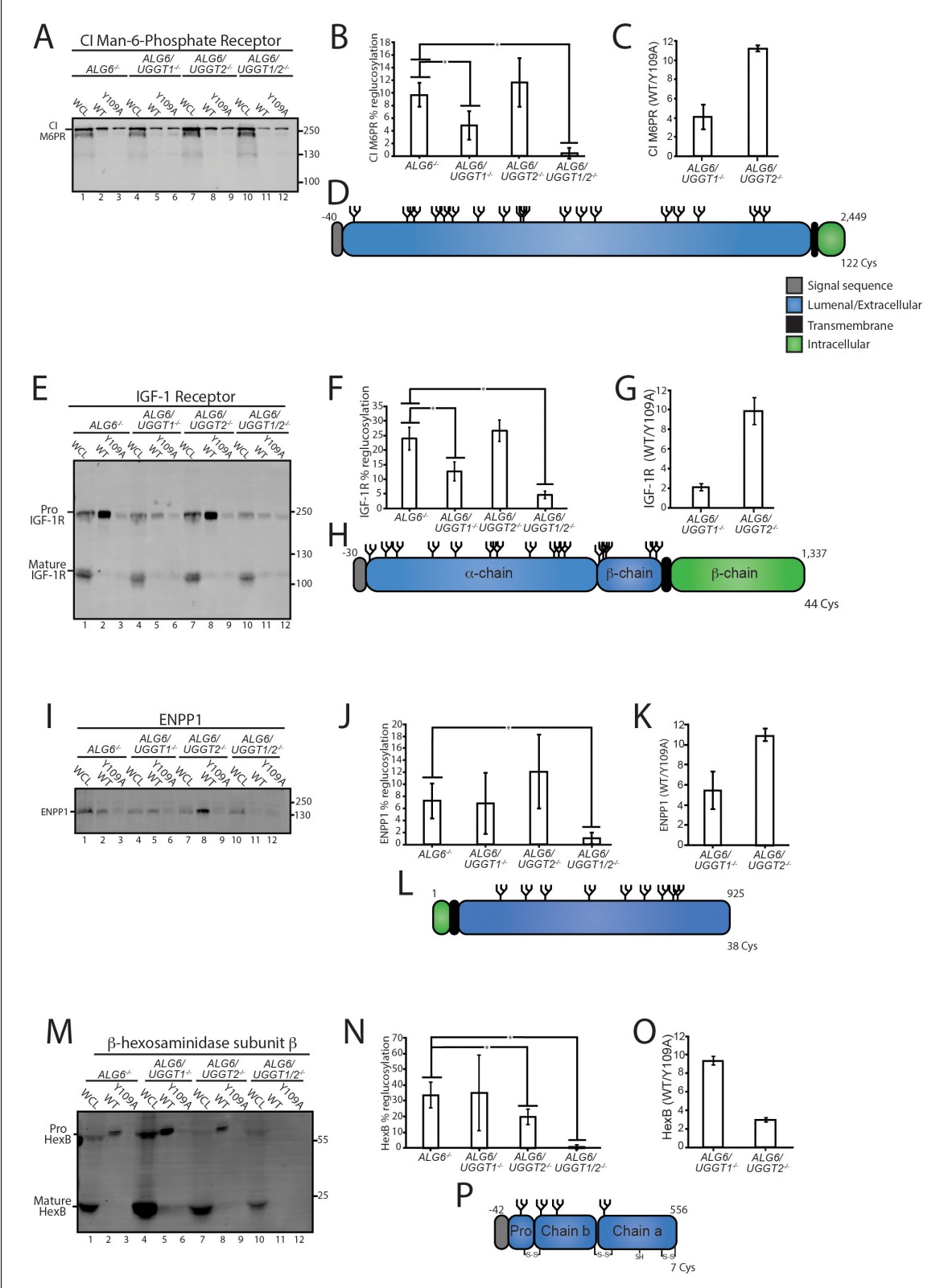

**Figure 3.** Validation of select reglucosylation substrates. (**A**) The designated cell lines were lysed and split into whole cell lysate (WCL, 10%) or affinity purification by GST-CRT-WT or GST-CRT-Y109A and imaged by immunoblotting against the CI Man-6-Phosphate receptor. Data is representative of three independent experiments with quantification shown in panel (**B**). Quantifications were calculated by subtracting the value of protein in the Y109A lane from the value of protein in the associated wild-type (WT) lane, divided by the value of protein in the associated WCL lane and multiplied by 100.

*Figure 3 continued on next page*

*Figure 3 continued*

Error bars represent the standard deviation. Asterisks denote a p-value of less than 0.05. (C) Tandem mass tag (TMT) mass spectrometry quantification of CI Man-6-Phosphate receptor reglucosylation from *ALG6/UGGT1⁻/⁻* cells (*Figure 2B*) and *ALG6/UGGT2⁻/⁻* cells (*Figure 2A*). (D) Cartoon representation of CI Man-6-Phosphate receptor with N-glycans (branched structures), the signal sequence (gray), luminal/extracellular domain (blue), transmembrane domain (black), and intracellular domain (green) depicted. Number of amino acids and Cys residues are indicated. (E) Reglucosylation of IGF-1R, conducted as previously described above. Pro IGF-1R and mature IGF-1R are both observed due to proteolytic processing. Data are representative of three independent experiments with quantification displayed in (F). (G) TMT mass spectrometry quantification of IGF-1R from *Figure 2A and B*, as previously described. (H) Cartoon depiction of IGF-1R. (I) The reglucosylation of ENPP1 shown with quantification displayed in J. (K) TMT mass spectrometry quantification of ENPP1 from *Figure 2A and B* with cartoon depiction of ENPP1 in L. (M) Reglucosylation of β-hexosaminidase subunit β, conducted as previously described with quantifications displayed in N and TMT mass spectrometry quantification of β-hexosaminidase subunit β from *Figure 2A and B* in O with a cartoon depicting β-hexosaminidase subunit β in P.

The online version of this article includes the following source data and figure supplement(s) for figure 3:

**Source data 1.** Quantifications for reglucosylation validations.
**Figure supplement 1.** mRNA expression of lysosomal preferential UDP-glucose:glycoprotein glucosyltransferase (UGGT)2 substrates.
**Figure supplement 2.** UPR induction in knockout cell lines.

approach did not require a high number of monoglucosylated glycans for GST-CRT affinity isolation but substrates possessing multiple reglucosylated sites are likely affinity isolated more efficiently by the GST-CRT pull downs.

The ER maintains an oxidizing environment that supports the formation of disulfide bonds. Complex folding pathways can involve the engagement of oxidoreductases, such as the calnexin/calreticulin-associated oxidoreductase ERp57, to catalyze disulfide bond formation and isomerization (*Margittai and Sitia, 2011*; *Kozlov and Gehring, 2020*). The most common number of Cys in UGGT substrates was similar to the N-glycoproteome Cys content (2 Cys, *Figure 4C*). However, the median number of Cys residues for the N-glycoproteome (11 Cys) is smaller than that found in *ALG6⁻/⁻* cells (16 Cys) and *ALG6/UGGT2⁻/⁻* cells (13 Cys). In contrast, a median of nine Cys was observed for UGGT2 substrates. Therefore, UGGT1 appears to display a slight preference for proteins with high Cys content, when compared to the N-glycoproteome and UGGT2 substrates.

UGGT1 or UGGT2 substrates displayed similar pI distributions with pIs predominantly near a pH of 6.0, while a second smaller cluster centered around a pH of 8.5. Interestingly, a pronounced low-density region was observed at pH 7.9 under all conditions, presumably due to the instability of proteins with pIs of a similar pH to that of the ER. The N-glycoproteome displayed a more bimodal distribution with significant population of both acidic and basic pIs (*Figure 4D*). These results suggest that both UGGT1 and UGGT2 preferentially engage proteins with low pIs.

The predicted topologies of the substrates of the UGGTs and the N-glycoproteome were also analyzed. Approximately 70% of the N-glycoproteome is comprised of membrane proteins, with half of these membrane proteins possessing multiple transmembrane domains, followed by single membrane pass proteins with a type I orientation (a third) with the remainder being type II membrane proteins (*Figure 4E*). A total of 43% of UGGT substrates in *ALG6⁻/⁻* cells contained a transmembrane domain with the vast majority of these substrates having their C-terminus localized to the cytosol in a type I orientation, while two substrates possessed the reverse type II orientation and a single multi-pass membrane substrate (NPC1) was identified. When the UGGTs were considered separately, about half of the UGGT1 substrates (*ALG6/UGGT2⁻/⁻* cells) possessed at least one transmembrane domain, with 70% of these membrane proteins being in the type I orientation, a quarter in a type II orientation, and two being multi-pass proteins (NPC1 and scavenger receptor class B member 1 [SR-BI]). In contrast to UGGT1, the majority of UGGT2 substrates were soluble proteins (72%) with the breakdown of remaining transmembrane proteins being similar to that of UGGT1 with the majority being type I membrane proteins. The preference of UGGTs for type I transmembrane proteins is likely caused by their larger luminal-exposed domains and N-glycan numbers compared to multi-pass membrane proteins (*Figure 4F and G*). Notably, substrates of the UGGTs had larger luminal domains than the membrane proteins of the N-glycoproteome, though especially for the multi-pass membrane proteins (*Figure 4F*). Furthermore, while the pIs of type II and polytopic membrane proteins were bimodal, they were overall more basic, which appears to be a property disfavored by UGGT substrates (*Figure 4H*). Overall, these results show that UGGT1 efficiently modifies both soluble and membrane associated proteins, while UGGT2 strongly favors soluble substrates.

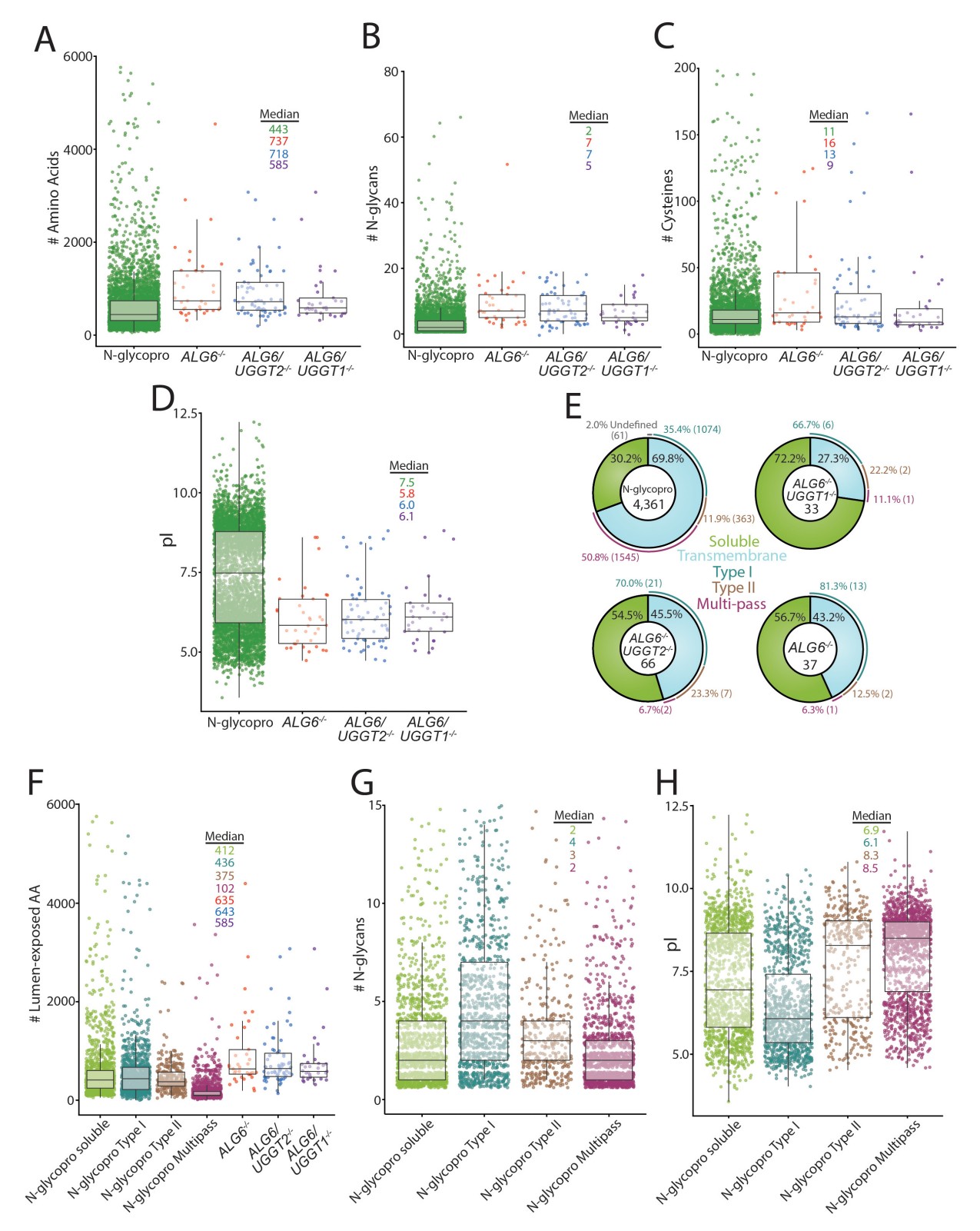

**Figure 4.** Analysis of substrates of the UDP-glucose:glycoprotein glucosyltransferase (UGGT)s and the N-glycoproteome. (**A**) Amino acid lengths of each protein in the indicated data sets were visualized by scatter plot overlaid with a box and whisker plot. Amino acid number was obtained via UniprotKB. All scatter plots with box and whisker plots were generated using R and the ggplot package. The number of N-glycans (**B**) or Cys residues (**C**) for each protein in the indicated data sets was visualized by scatter plot overlaid with a box and whisker plot with the numbers determined using

*Figure 4 continued on next page*

*Figure 4 continued*

their UniprotKB annotation. (D) The isoelectric point (pI) values for each protein in the indicated data sets was visualized by scatter plot overlaid with a box and whisker plot. The pI values were obtained via ExPASy theoretical pI prediction. (E) The computationally predicted N-glycoproteome and the indicated reglucosylation substrates were determined as either soluble or transmembrane using UniprotKB annotations. The transmembrane portion of each data set was then analyzed for type I, type II, or multi-pass topology using the associated UniprotKB annotation. Proteins that were annotated by UniprotKB as transmembrane but lacked topology information were labeled as undefined. (F) The computationally determined N-glycoproteome was separated into soluble, type I, type II, and multi-pass transmembrane proteins using UniprotKB annotations. Luminally exposed amino acids were computationally determined using UniprotKB annotations for each subset of the N-glycoproteome and each indicated reglucosylation substrate data set. The resulting data was visualized by scatter plot overlaid with a box and whisker plot. (G) The indicated N-glycoproteome subsets were analyzed for N-glycan content using UniprotKB annotation and visualized by scatter plot overlaid with a box and whisker plot, as described. (H) The indicated N-glycoproteome subsets were analyzed for predicted pI using ExPASy theoretical pI prediction and visualized by scatter plot overlaid with a box and whisker plot.

The online version of this article includes the following source data for figure 4:

**Source data 1.** Characteristics of the N-glycoproteome.

## Efficient IGF-1R trafficking requires lectin chaperone engagement

A number of natural substrates of the UGGTs were identified using a glycoproteomics approach with gene edited cell lines. As reglucosylation by the UGGTs can direct multiple rounds of lectin chaperone binding, the necessity for reglucosylation to support the efficient maturation of a reglucosylated substrate was investigated. IGF-1R is proteolytically processed in the *trans*-Golgi by proprotein convertases including furin, facilitating the monitoring of IGF-1R trafficking from the ER to the Golgi (*Lehmann et al., 1998*). The requirement for lectin chaperone binding and reglucosylation to aid IGF-1R trafficking was analyzed.

Initially, cells were treated without or with the inhibitor of α-glucosidases I and II, DNJ, to accumulate IGF-1R in the triglucosylated state to bypass entry into the calnexin/calreticulin binding cycle (*Hammond and Helenius, 1994*; *Hebert et al., 1995*). At steady state as probed by immunoblotting of cell lysates, IGF-1R accumulated in the ER localized pro form relative to the mature form after DNJ treatment (*Figure 5A*), resulting in a 19% decrease in the level of the *trans*-Golgi processed mature protein (*Figure 5B*). This indicated that the lectin chaperone binding cycle helps support efficient IGF-1R trafficking.

There are two modes for engaging the lectin chaperone cycle: initial binding, which can potentially commence co-translationally for glycoproteins such as IGF-1R that have N-glycans located at their N-terminus through their trimming of the terminal two glucoses by glucosidases I and II; or by rebinding, which is directed by the reglucosylation of unglucosylated species by the UGGTs (*Caramelo and Parodi, 2015*; *Lamriben et al., 2016*). The contribution of each mode of monoglucose generation for the proper trafficking of IGF-1R was analyzed.

IGF-1R maturation was investigated in *ALG6*[-/-] cells as in these cells the N-glycan transferred to the nascent substrate is non-glucosylated, leading to a lack of initial glucosidase trimming mediated lectin chaperone binding. Reglucosylation by the UGGTs is required for lectin chaperone binding in *ALG6*[-/-] cells. Similar to DNJ treatment in WT cells, *ALG6*[-/-] cells demonstrate a 20% decrease in mature IGF-1R relative to the pro form at steady state (*Figure 5C*, lanes 1 and 3, and *Figure 5D*). As hypoglycosylation can occur in a substrate dependent manner in *Alg6*[-/-] cells (*Shrimal and Gilmore, 2015*), the mobility of IGF-1R with and without N-glycans (PNGase F treated) was monitored by comparing the mobility of IGF-1R by SDS-PAGE and immunoblotting of WT and *ALG6*[-/-] cell lysates. IGF-1R, and similarly, CI-M6PR, appeared to be fully glycosylated, while HexB migrated faster when synthesized in *ALG6*[-/-] cells likely due to hypoglycosylation (*Figure 5C* and *Figure 2—figure supplement 3*). To confirm that the pro form of IGF-1R represented ER localized protein rather than protein trafficked out of the ER but not processed by proprotein convertases, IGF-1R from WT and *Alg6*[-/-] cells was treated with the endoglycosidase EndoH. As EndoH cleaves high-mannose glycans that are preferentially present in the ER or early Golgi, an increase in mobility by SDS-PAGE suggests ER localization. In both WT and *Alg6*[-/-] cells, Pro IGF-1R was observed to be EndoH sensitive, while mature IGF-1R was found to be largely EndoH resistant (*Figure 5C*, lanes 2 and 5), suggesting the accumulation of pro IGF-1R in *Alg6*[-/-] cells represents impaired ER trafficking rather than impaired processing in the *trans*-Golgi. Altogether, these steady state results suggest that lectin chaperone binding is important for efficient IGF-1R maturation.

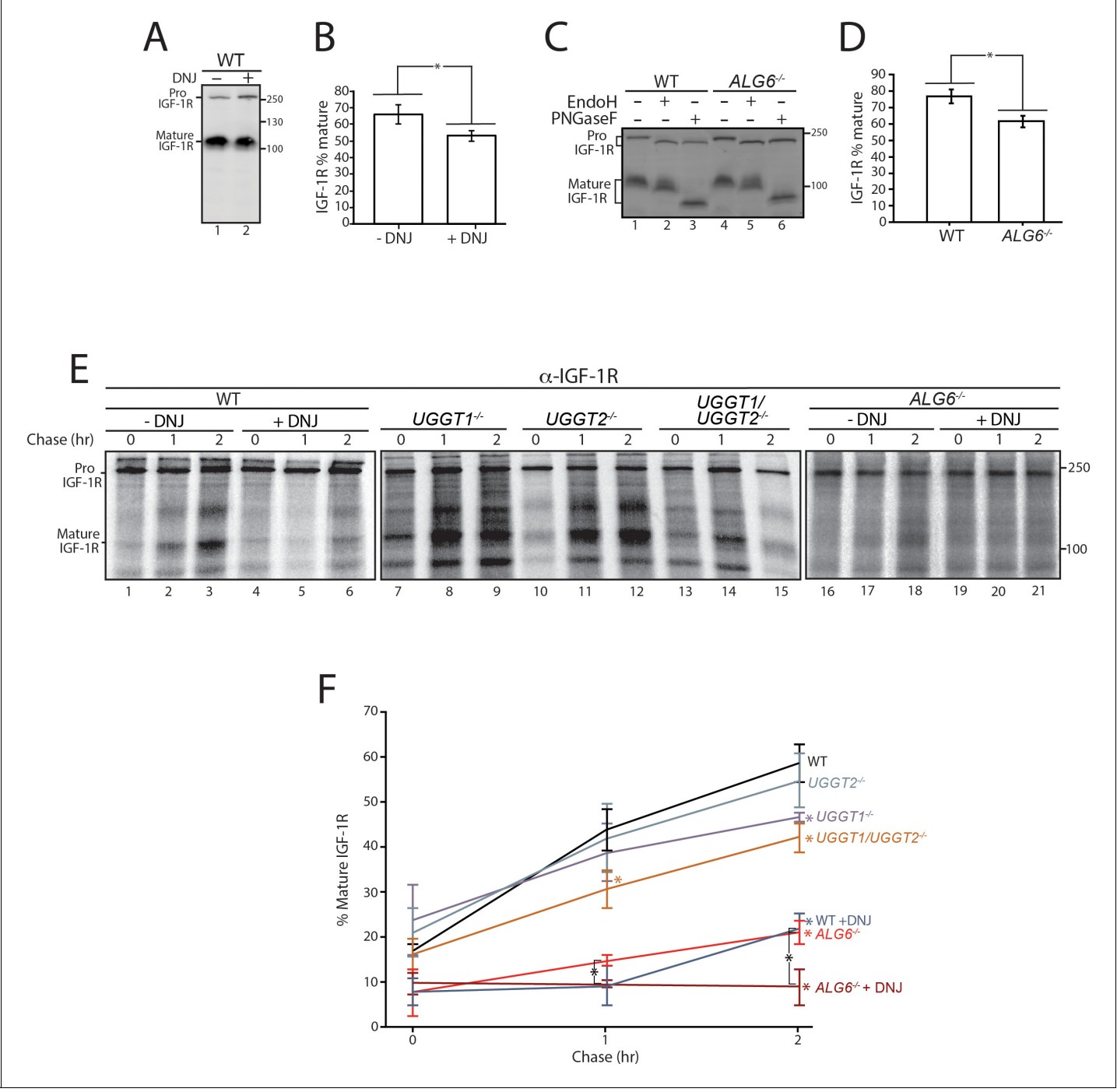

**Figure 5.** Calnexin/calreticulin cycle role for IGF-1R trafficking. (**A**) Wild-type HEK293-EBNA1-6E cells treated without or with deoxynojirimycin (DNJ; 500 µM) for 12 hr were lysed and whole cell lysate samples were resolved by reducing 9% SDS-PAGE and imaged by immunoblotting against IGF-1R. Data are representative of three independent experiments with quantification shown in (**B**). Percent of IGF-1R mature was calculated by dividing the amount of mature protein by the total protein in each lane. Errors bars represent standard deviation. Asterisk denotes a p-value of less than 0.05. (**C**) The indicated cell lines were lysed in RIPA buffer. Samples were split evenly between non-treated and PNGaseF or EndoH treated. Samples were visualized by immunoblotting against IGF-1R and data are representative of three independent experiments with quantification displayed in (**D**). (**E**) Indicated cells were treated without or with DNJ, pulsed with [35S]-Met/Cys for 1 hr and chased for the indicated times. Cells were lysed and samples were immunoprecipitated using anti-β IGF-1R antibody and resolved by reducing SDS-PAGE and imaged by autoradiography. Data are representative of three independent experiments with quantification shown in (**F**).

The online version of this article includes the following source data for figure 5:

**Source data 1.** Quantifications for IGF-1R trafficking puse chase.

As steady state results can be impacted by changes in protein synthesis and turnover, a radioactive pulse-chase approach was used to follow protein synthesized during a 1 hr [$^{35}$S]-Met/Cys pulse interval followed by chasing for up to 2 hr under non-radioactive conditions. Pulse-chase experiments are generally performed with overexpressed tag constructs to accumulate and isolate sufficient protein for monitoring. Here endogenous IGF-1R was isolated by immunoprecipitation with anti-IGF-1R antibodies and analyzed by SDS-PAGE and autoradiography to determine the percent of IGF-1R that was properly processed to its mature form in the *trans*-Golgi. IGF-1R was found to traffic efficiently out of the ER and to the Golgi in WT cells as 59% of the total protein after a 2 hr chase was mature IGF-1R (*Figure 5E*, lanes 1–3, and F). When lectin chaperone binding was inhibited by treatment with DNJ, mature IGF-1R was diminished to 22%, underscoring the importance of lectin chaperone binding (*Figure 5E*, lanes 4–6, and F).

To delineate the contributions of early compared to late lectin chaperone binding, IGF-1R trafficking was followed in gene edited cells that control the mechanisms for lectin chaperone engagement. A single early round of lectin chaperone binding will be permitted in the absence of both UGGTs or rebinding would only be directed by the UGGT present with knockouts of a single UGGT. Alternatively, early lectin chaperone binding as dictated by glucosidase trimming will be absent in the *ALG6*$^{-/-}$ cells where lectin chaperone binding is directed solely through glucosylation by the UGGTs. Monitoring the trafficking of IGF-1R in these cells will allow us to determine the contributions of the different steps in the lectin chaperone binding cycle for proper IGF-1R maturation.

When both UGGTs were absent in *UGGT1/2*$^{-/-}$ cells, the percent of mature IGF-1R after 2 hr of chase decreased to 42%. In agreement with early glycoproteomics and affinity isolation results showing IGF-1R was largely a UGGT1 substrate, UGGT2 knockout alone had little influence on IGF-1R trafficking while the knocking out of UGGT1 supported IGF-1R trafficking similar to the double UGGT deletion (*Figure 5E,* lanes 7–15, and F). These results support a role for UGGT1 in optimizing IGR-1R trafficking.

To determine the importance of early chaperone binding directed by the glucosidases, IGF-1R trafficking was monitored in *ALG6*$^{-/-}$ cells that support reglucosylation but lack the ability for early binding to the lectin chaperones as directed by glucosidase trimming of the triglucosylated species. In *ALG6*$^{-/-}$ cells, the percent of mature IGF-1R was significantly decreased to 21%, indicative of an important contribution of the initial round of lectin binding, as was suggested by steady state data (*Figure 5C*). The addition of DNJ to *ALG6*$^{-/-}$ cells would be expected to trap IGF-1R in a monoglucosylated state after glucosylation, allowing the effect of prolonged interaction with the lectin chaperones to be observed. Under this condition, IGF-1R was strongly retained in the ER with no increase observed in the level of mature IGF-1R observed even after 2 hr of chase (*Figure 5E*, lanes 16–21, and F). Altogether these results demonstrate that while early (glucosidase-mediated) and late (UGGT-mediated) lectin chaperone binding contribute to the efficient trafficking from the ER and subsequent Golgi processing of IGF-1R, early lectin chaperone binding appears to be most critical for supporting proper IGF-1R maturation.

## Discussion

As lectin chaperone binding is directed by the covalent modification of substrates by the UGGTs, the identification of *bona fide* substrates of the UGGTs is central to understand the impact the lectin chaperone network has on cellular homeostasis. Features of proteins alone cannot accurately predict which chaperones will be required for efficient folding and quality control (*Adams et al., 2019b*). Previous studies involving the UGGTs have focused mainly on the overexpression of biasedly selected substrates or using purified proteins, providing uncertain biological relevance (*Ritter and Helenius, 2000*; *Taylor et al., 2003*; *Caramelo et al., 2004*; *Soldà et al., 2007*; *Pearse et al., 2008*; *Ferris et al., 2013*; *Tannous et al., 2015*). Here we used a quantitative glycoproteomics-based strategy to identify 71 natural cellular substrates of the UGGTs. When compared to the N-glycoproteome that represents the total population of potential substrates (4361 N-glycoproteins in human cells), the UGGTs favored the modification of more complex, multidomain proteins with large numbers of N-glycans. These results are in agreement with the common requirement of chaperones for the proper folding of more complex proteins (*Balchin et al., 2016*; *Balchin et al., 2020*). The lectin chaperone system is part of the robust chaperone network necessary to promote the efficient

folding and quality control of substrates and mitigate harmful misfolding events that are associated with a large range of pathologies.

The discovery of 33 UGGT2 cellular substrates provides the first evidence of intact UGGT2 acting as a quality control factor in cells (*Figure 2B*). Previous work demonstrated that UGGT2 is enzymatically active against chemically engineered glycosylated substrates using purified components or when the catalytic domain of UGGT2 was appended to the folding sensor domain of UGGT1 (*Arnold and Kaufman, 2003*; *Takeda et al., 2014*). The lower number of UGGT2 substrates compared to UGGT1 (66 substrates) is likely due, at least in part, to UGGT2 being expressed at a fraction of the level of UGGT1 (~4% in HeLa cells *Itzhak et al., 2016*). Of special note is the preference of UGGT2 for lysosomal substrates as eight of the nine preferential UGGT2 substrates are lysosomal proteins (*Figure 2E*). The preferential UGGT2 substrates are all soluble proteins, while half of the preferential UGGT1 substrates contained transmembrane domains indicative of a further preference of UGGT2 for soluble proteins (*Figure 2E*). Given the preference of UGGT2 for soluble lysosomal proteins, it would be of interest in future studies to examine lysosomes in *UGGT2⁻/⁻* cells or mice as a number of the UGGT2 substrates are associated with lysosomal storage diseases including metachromatic leukodystrophy (arylsulfatase A), Fabry (alpha-galactosidase A), Sandhoff (β-hexosaminidase subunit β), and Schindler (α-N-acetylgalactosaminidase) diseases (*Mahuran, 1999*; *Cesani et al., 2016*; *Ferreira and Gahl, 2017*).

UGGT1 serves as the predominant ER glycoprotein quality control sensor. While overall the 66 UGGT1 substrates are evenly distributed between soluble and membrane proteins, the majority of the most efficiently reglucosylated proteins are membrane proteins (*Figure 2E*). Seventy percent of the membrane proteins modified by UGGT1 are in the type I orientation possessing luminal N-glycosylated domains of significant length. Only two substrates of the UGGTs are multi-pass membrane proteins (NPC1 and SR-BI). In contrast to most polytopic membrane proteins that have little exposure to the ER lumen (*Figure 4F*), both NPC1 and SR-BI have large heavily glycosylated luminal domains. The enrichment of UGGT1 for transmembrane proteins may be influenced through a weak association with the ER membrane or a general slower and more complex folding process for membrane proteins that provides a longer window for modification. As a majority of the UGGT substrates are found in oligomeric complexes, the UGGTs might also exhibit a preference for unassembled subunits to help ensure proper protein assembly in the ER.

An important question to ask is what is the basis for the differing substrate specificities of UGGT1 and UGGT2? They display sequence identities that are high within the catalytic domains (83% identical) and lower in their folding sensor domains (49%) (*Arnold and Kaufman, 2003*). This sequence disparity within the folding sensor domain may drive altered substrate selection. In addition, UGGT1 and UGGT2 may reside in separate subdomains within the ER, and this could contribute to substrate accessibility. The CLN6/CLN8 transmembrane complex appears to recognize lysosomal proteins within the ER for COPII packaging in support of a possible mechanism of lysosomal substrate selection (*Bajaj et al., 2020*). An additional possibility addressed was that the level of expression of the lysosomal proteins identified as UGGT2 substrates may be augmented in *ALG6/UGGT1⁻/⁻* cells. However, only the mRNA expression level of β-hexosaminidase subunit β was increased relative to *ALG6⁻/⁻* or WT cells, as supported by immunoblot data (*Figure 3M*), with the remaining preferential UGGT2 lysosomal substrates displaying no significant change in mRNA expression levels (*Figure 3—figure supplement 1*). The increased expression of β-hexosaminidase subunit β in *ALG6/UGGT1⁻/⁻* cells may be attributed to induction by UPR, as in these cells a slight induction primarily through the ATF6 branch of the UPR was observed (*Figure 3—figure supplement 2*). Further studies will be required to understand the varying selectivities of the UGGTs.

With some 4350 possible N-glycosylated proteins as potential UGGT substrates, why were only 71 proteins identified as substrates of the UGGTs? First, many proteins are expected to fold in a chaperone independent manner, especially small, simple proteins. Second, our stringent isolation approach prioritized high quality substrates with at least a threefold induction for GST-CRT/GST-CRT-Y109A binding. Third, the profile of reglucosylated substrates is likely cell-type dependent with additional substrates expected to be identified in cell types with heavy secretory pathway loads such as pancreatic cells or hepatocytes, compared to the kidney line used here. Fourth,~1500 proteins of the N-glycoproteome are multi-pass transmembrane proteins (*Figure 4E*). This class of protein was strongly de-enriched as substrates of the UGGTs, likely due to their limited luminal exposure, minimal N-glycan content, and difficulties in isolating hydrophobic polytopic membrane proteins

(*Figure 4F and G*). This reduces the pool of favored substrates by one-third. Finally, the monoglucosylated protein isolation procedure that relies on CST-CRT pull downs is likely influenced by the number of monoglucosylated sites on a protein. While multiple UGGT substrates with two N-glycans were identified, suggesting extensive glycosylation is not a requirement, the number of monoglucosylated glycans on a substrate is expected to have an impact on the efficiency of substrate isolation and identification. We hope to identify specific sites of reglucosylation in future studies. These results would demonstrate if reglucosylation occurs on multiple sites or a small number of sites.

Protein expression levels are also expected to play some role in substrate identification. However, it does not appear to be a major determining factor as multiple strong substrates were expressed at or below an average protein level for the N-glycoproteome and no correlation between mRNA expression level and the TMT mass spectrometry fold increase for the GST-CRT/GST-CRT-Y109A fraction was observed (*Figure 2—figure supplement 2*). It would be of interest to determine if proteotoxic stress would increase levels and the range of reglucosylated substrates as both the pool of non-native proteins and the amount of the UPR-induced substrates of the UGGTs would be expected to increase.

As carbohydrate binding can be dictated initially by glucosidase trimming followed by additional later rounds of binding dictated by UGGT reglucosylation, it is of importance to understand which stage of the binding cycle contributes most significantly to proper protein maturation and cell homeostasis. N-glycans in *Sacchromyces cerevisiae* and other single cell species are transferred post-translationally as they are missing the OST isoform subunit that interacts with the Sec61 translocon and supports early co-translational modification (*Ruiz-Canada et al., 2009*; *Shrimal et al., 2019*). A second OST isoform appears in multicellular organisms that is translocon-associated (*Braunger et al., 2018*; *Ramírez et al., 2019*). In addition, reglucosylation activity was first observed in single cell parasites of *Trypanosoma cruzi* where glycans are transferred as $Man_9GlcNAc_2$ moieties thereby bypassing the initial glucosidase initiated binding step observed in metazoans (*Parodi and Cazzulo, 1982*). These seminal *T. cruzi* studies from Parodi and colleagues that first discovered the (re)glucosylation activity, later attributed to UGGT1, were the inspiration for the development of the experimental $ALG6^{-/-}$ system used in this study to isolate substrates of the UGGTs. Conservation analysis of glycosylation and the lectin chaperone pathway suggests that reglucosylation supporting the quality control function of the calnexin cycle evolved prior to its role in assisting in earlier folding events.

Using CRISPR edited cell lines, the contributions of the various steps for chaperone binding engagement for the UGGT1 substrate IGF-1R was experimentally explored as its processing in the Golgi provided a robust Golgi trafficking assay. Furthermore, IGF-1R is a target in cancer biology as it is important for cell growth (*Sell et al., 1994*; *Desbois-Mouthon et al., 2006*; *Chng et al., 2006*; *King et al., 2014*; *Mutgan et al., 2018*). When binding to the lectin chaperones was blocked in WT cells by glucosidase inhibition with DNJ treatment, supporting the production of triglucosylated trapped species, the percent of processed IGF-1R strongly decreased compared to untreated cells. This demonstrated a requirement of lectin chaperone engagement for the efficient maturation, trafficking, and processing of IGF-1R. In $UGGT1/2^{-/-}$ cells, IGF-1R can enter the first round of glucosidase-mediated binding to the lectin chaperones but rebinding directed primarily by UGGT1-mediated reglucosylation cannot occur (*Figures 5E and F* and *6*). This led to a reduced efficiency in the accumulation of mature IGF-1R. The first round of lectin chaperone binding is bypassed in $ALG6^{-/-}$ cells as the N-glycans transferred to proteins do not contain glucoses (*Figure 6*). Therefore, only the rebinding events mediated by reglucosylation takes place. More strikingly in $ALG6^{-/-}$ cells, this led to a dramatic reduction in IGF-1R processing at a greater level than in $UGGT1/2^{-/-}$ cells, indicating the first round of binding to the lectin chaperones was most critical for IGF-1R maturation. The addition of DNJ in $ALG6^{-/-}$ cells supported the trapping of reglucosylated side chains and severely reduced Golgi processing, suggesting that reglucosylation-mediated persistent interaction with the lectin chaperones delays IGF-1R exit from the ER.

$ALG6^{-/-}$ cells permitted the trapping of substrates glucosylated by the UGGTs. These cells are also expected to support the enhancement of glucosylation of glycoproteins that are more reliant upon early lectin chaperone intervention. As observed for IGF-1R, the lack of early intervention of the lectin chaperones directed by glucosidase trimming might lead to misfolding, thereby creating a better substrate for the UGGTs. The use of the cell lines lacking the ability to initiate lectin chaperone binding by the glucosidase trimming ($Alg6^{-/-}$) or UGGT reglucosylation ($UGGT^{-/-}$ cells) provides

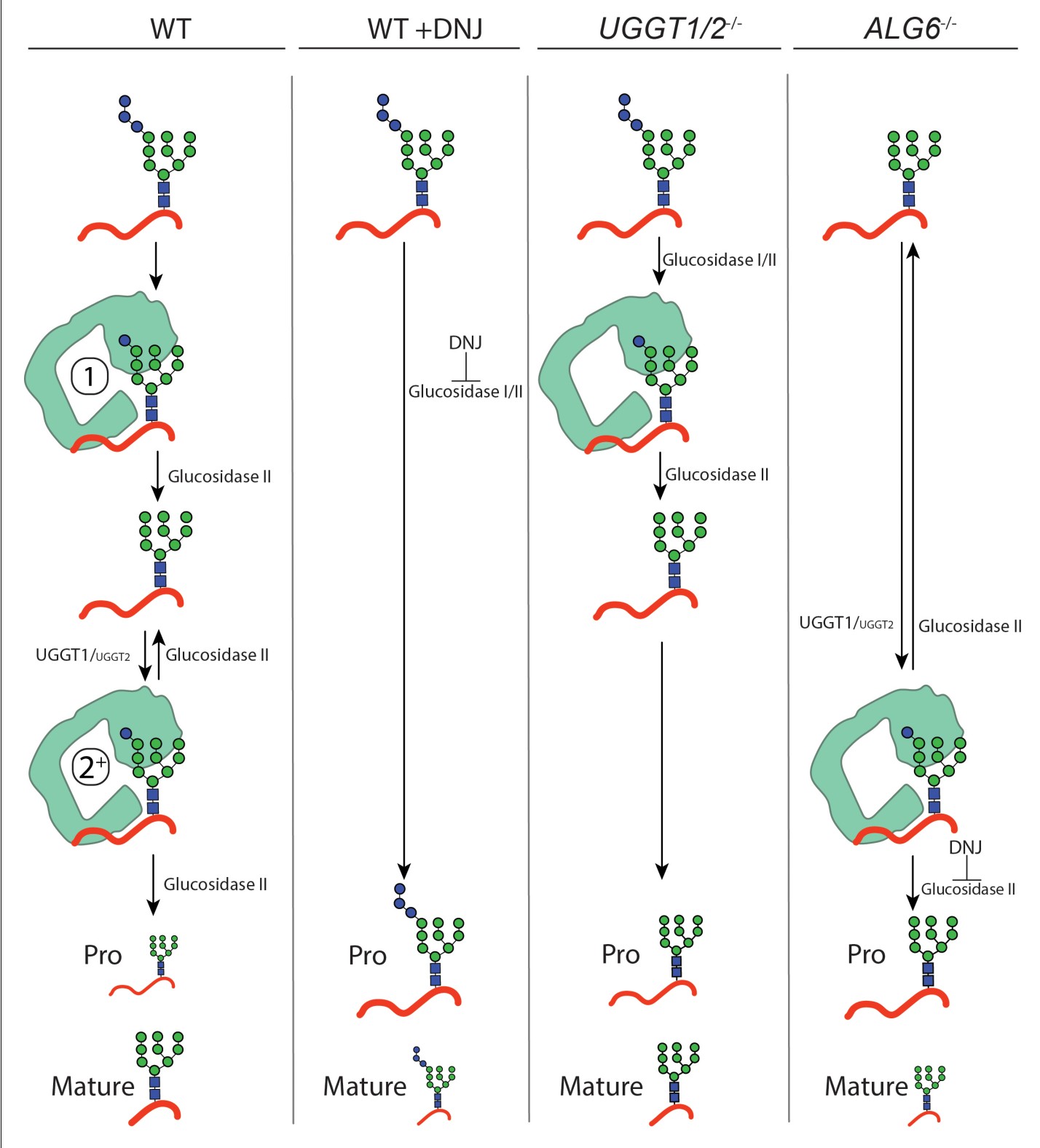

**Figure 6.** Model for IGF-1R engagement by the lectin chaperone cycle. In wild-type (WT) cells, N-glycans with three terminal glucoses are appended to IGF-1R. Trimming of two terminal glucoses by glucosidases I/II generates a monoglucosylated protein that supports an initial round of interaction with calreticulin (calnexin not shown, denoted by a 1). Trimming of the final glucose by glucosidase II yields a non-glucosylated N-glycan. If recognized as non-native primarily by UDP-glucose:glycoprotein glucosyltransferase (UGGT)1, and to a lesser extent UGGT2, IGF-1R may then be reglucosylated, supporting a second round of interaction with calreticulin (denoted by a 2+). Multiple rounds of trimming, reglucosylation, and binding to calnexin or

*Figure 6 continued on next page*

*Figure 6 continued*

calreticulin can occur until proper folding and trafficking. Under this system, IGF-1R is efficiently trafficked from the ER and mature IGF-1R accumulates. When glucosidase I/II activity is inhibited by treatment with deoxynojirimycin (DNJ) in WT cells, all rounds of binding to the lectin chaperones are ablated and IGF-1R is retained in the ER, yielding primarily pro IGF-1R. In *UGGT1/2*$^{-/-}$ cells, initial binding to calnexin or calreticulin directed by glucosidases I/II trimming is maintained but rebinding via reglucosylation does not occur. Under this system, IGF-1R is inefficiently trafficked from the ER. In *ALG6*$^{-/-}$ cells, N-glycans are transferred without glucoses, eliminating the initial round of binding to calnexin or calreticulin by glucosidases trimming. Only the second round of binding is supported by UGGT1, and to a lesser extend UGGT2, mediated reglucosylation. Upon treatment with DNJ, reglucosylated IGF-1R may persistently interact with the lectin chaperones resulting in ER retention.

a platform to delineate which part of the lectin chaperone binding cycle has the greatest influence of glycoprotein maturation and trafficking.

Understanding the proteins that interact with or rely on chaperone systems will advance our understanding of protein homeostasis (*Houry et al., 1999*; *Kerner et al., 2005*). Large multi-domain proteins such as IGF-1R and many of the other substrates of the UGGTs have apparently evolved to utilize the lectin chaperone system to help direct their complex folding trajectories. The co-evolution of chaperones and their substrates has led to the expansion of the complexity of the proteome for multicellular organisms (*Balchin et al., 2016*; *Rebeaud et al., 2020*). The large group of substrates of the UGGTs identified here represents glycoproteins that utilize multiple rounds of lectin chaperone engagement for proper maturation and are likely more prone to misfold under stress. Future studies will determine if this increased vulnerability makes these substrates more susceptible to misfold under disease conditions where cell homeostasis is challenged.

# Materials and methods

### Key resources table

| Reagent type (species) or resource | Designation | Source or reference | Identifiers | Additional information |
|---|---|---|---|---|
| Strain, strain background (*Escherichia coli*) | Top10 | Thermo Fisher | Cat# C404003 | Chemically competent |
| Cell line (*H. sapiens*) | Hek293-EBNA1-6E | This paper | (RRID:CVCL_HF20) | Experimental results |
| Cell line (*H. sapiens*) | Hek293-EBNA1-6E *ALG6*$^{-/-}$ | This paper | | Experimental results |
| Cell line (*H. sapiens*) | Hek293-EBNA1-6E *ALG6/UGGT1*$^{-/-}$ | This paper | | Experimental results |
| Cell line (*H. sapiens*) | Hek293-EBNA1-6E *ALG6/UGGT2*$^{-/-}$ | This paper | | Experimental results |
| Cell line (*H. sapiens*) | Hek293-EBNA1-6E *ALG6/UGGT1/2*$^{-/-}$ | This paper | | Experimental results |
| Cell line (*H. sapiens*) | Hek293-EBNA1-6E *UGGT1*$^{-/-}$ | This paper | | Experimental results |
| Cell line (*H. sapiens*) | Hek293-EBNA1-6E *UGGT2*$^{-/-}$ | This paper | | Experimental results |
| Cell line (*H. sapiens*) | Hek293-EBNA1-6E *UGGT1/2*$^{-/-}$ | This paper | | Experimental results |
| Antibody | IGF-1 receptor β (D23H3) (rabbit monoclonal) | Cell Signaling | Cat # 9750 | WB (1:1000) IP (1:1000) |
| Antibody | IGF-IIR/CI-M6PR (D3V8C) (rabbit monoclonal) | Cell Signaling | Cat# 14364 | WB (1:1000) |

*Continued on next page*

*Continued*

| Reagent type (species) or resource | Designation | Source or reference | Identifiers | Additional information |
|---|---|---|---|---|
| Antibody | β-hexosaminidase subunit β (EPR7978) (rabbit monoclonal) | Abcam | Cat# (ab140649) | WB (1:500) |
| Antibody | BiP (C50B12) (rabbit monoclonal) | Cell Signaling | Cat# 3177 | WB (1:1000) |
| Antibody | ENPP1 (N2C2) (rabbit polyclonal) | Genetex | Cat# GTX103447 | WB (1:500) |
| Antibody | UGGT1 (rabbit polyclonal) | Genetex | Cat# GTX66459 | WB (1:1000) |
| Antibody | Glyceraldehyde 3-Phosphate (mouse monoclonal) | Millipore Sigma | Cat# (MAB374) | WB (1:1000) |
| Recombinant DNA reagent | pGEX-3X-GST-CRT (plasmid) | *Baksh and Michalak, 1991* | | Available from the Hebert lab upon request |
| Recombinant DNA reagent | pGEX-3X-GST-CRT-Y109A (plasmid) | This paper | | Available from the Hebert lab upon request |
| Recombinant DNA reagent | gh260 | *Narimatsu et al., 2018* | RRID:Addgene_106851 | gRNA for ALG6$^{-/-}$ |
| Recombinant DNA reagent | gh172 | *Narimatsu et al., 2018* | RRID:Addgene_106833 | gRNA for UGGT1$^{-/-}$ |
| Recombinant DNA reagent | gh173 | *Narimatsu et al., 2018* | RRID:Addgene_106834 | gRNA for UGGT2$^{-/-}$ |
| Recombinant DNA reagent | Cas9-GFP CAS9PBKS | *Lonowski et al., 2017* | RRID:Addgene_68371 | Cas9 for CRISPR-mediated knockout |
| Sequence-based reagent | CRT-Y109A_F | This paper | PCR primers | GGGGGCGGCG CCGTGAAGCT |
| Sequence-based reagent | CRT-Y109A_R | This paper | PCR primers | CCGGAAACAGCT TCACGTAGCCGC |
| Commercial assay or kit | TMT10plex, 0.8 mg | Thermo Fisher | Cat# 90110 | |
| Commercial assay or kit | TMT6plex, 0.8 mg | Thermo Fisher | Cat# 90061 | |
| Commercial assay or kit | BCA protein quantification kit | Pierce | Cat# 23227 | |
| Commercial assay or kit | C18 tips | Pierce | Cat # 87784 | |
| Commercial assay or kit | Quantitative colorimetric peptide assay | Pierce | Cat # 23275 | |

## Reagents

Antibodies used were: rabbit monoclonal IGF-1 receptor β (D23H3, Cell Signaling), rabbit monoclonal IGF-IIR/CI-M6PR (D3V8C, Cell signaling), rabbit monoclonal BiP (C50B12, Cell Signaling), rabbit monoclonal β-hexosaminidase subunit β (HEXB) (EPR7978, Abcam), rabbit polyclonal ENPP1 (N2C2, Genetex), rabbit polyclonal UGGT1 (GTX66459, Genetex), mouse monoclonal glyceraldehyde 3-

phosphate (MAB374, Millipore Sigma), and IRDye × anti-rabbit secondary (LiCor). All chemicals were purchased from Millipore-Sigma, except where indicated.

## Cell culture

HEK293-EBNA1-6E cells were employed and used as the parental line to create all CRISPR/Cas9 edited lines (*Tom et al., 2008*). Cells were cultured in DMEM (Sigma) supplemented with certified 10% fetal bovine serum (Gibco) at 37°C at 5% $CO_2$. Cells were tested for the presence of mycoplasma using a universal mycoplasma detection kit (ATCC, Cat # 30–012K).

## CRISPR/Cas9-mediated knock outs

HEK293-EBNA1-6E *ALG6$^{-/-}$*, *ALG6/UGGT1$^{-/-}$*, *ALG6/UGGT2$^{-/-}$*, *ALG6/UGGT1/UGGT2$^{-/-}$*, *UGGT1$^{-/-}$*, *UGGT2$^{-/-}$*, and *UGGT1/2$^{-/-}$* cells were generated via CRISPR/Cas9 using gRNA plasmids gh260, gh172, and gh173, and Cas9-GFP plasmid CAS9PBKS (*Lonowski et al., 2017*; *Narimatsu et al., 2018*). Plasmids gh260 (106851), gh172 (106833), gh173 (106834), and CAS9PBKS (68371) were from Addgene. Knockout cell lines were generated by co-transfecting HEK293-EBNA1-6E cells at 70% confluency in a 10 cm plate with 7 µg of both the associated gRNA and Cas9-GFP plasmid, using a 2.5 µg of PEI per 1 µg of plasmid. Cells were grown for 48 hr prior to trypsinization and collection. After trypsinization, cells were washed twice with sorting buffer (1% FBS, 1 mM EDTA, PBS) and resuspended in sorting buffer at approximately 1 million cells per milliliter. Cells were bulk separated using flow assisted cell sorting based on the top 10% of Cas9-GFP expressing cells (FACS Aria II SORP, Becton Dickinson and Company). Cells were then plated at 5,000, 10,000, and 20,000 cells per 10 cm plate in pre-conditioned DMEM media with 20% FBS. Colonies derived from a single cell were isolated using cell cloning cylinders (Bellco Glass), trypsinized from the plate, and further passaged. Knockouts were confirmed by immunoblotting and staining for UGGT1 or, where antibodies were not available, isolating genomic DNA using a genomic DNA isolation kit (PureLink genomic DNA mini kit, Thermo Fisher), PCR amplification of the genomic DNA region of interest, and insertion of genomic DNA into pcDNA3.1−. Plasmids were then sequenced for conformation (Genewiz).

## GST-CRT purification

The plasmid for pGEX-3X GST-CRT was from Prof. M. Michalak (University of Alberta). pGEX-3X GST-CRT-Y109A was generated by site-directed mutagenesis. GST-CRT was expressed in BL21 *E. coli* cells in LB medium containing ampicillin at 100 µg/ml. Cultures were grown at 37°C with shaking until an O.D. of $A_{600} = 0.6$. Protein expression was then induced by treating cultures with 8.32 mg/l IPTG for 2 hr. Cultures were centrifuged at 3000 g for 10 min. Cell pellets were lysed with cold lysis buffer (1 mM phenylmethylsulfonyl fluoride, 2% Triton X-100, PBS pH 7.4) and resuspended. Resuspended cells were lysed in a microfluidizer (110L, Microfluidics) at 18,000 psi for two passes. The cell lysate was centrifuged for 40 min at 8000 g at 4°C. Lysate was filtered through a 0.45 µm filter. Two milliliters bed volume glutathione sepharose beads (GE Lifesciences, Cat# GE17-0756-01) per liter of lysate was equilibrated in wash buffer (1% Triton X-100, 1 mM PMSF, PBS pH 7.4), added to cleared lysate, and rotated at 4°C for 3 hr. Beads were precipitated through centrifugation at 1000 g for 5 min at 4°C. The beads were washed twice in wash buffer. One milliliter of elution buffer (10 mM reduced glutathione, 1 mM PMSF, 50 mM Tris pH 8.5) was added to beads for resuspension and incubated for 5 min at 4°C. Beads were precipitated by centrifugation at 1000 g for 5 min 4°C. The eluate was collected and a total of six elutions were collected. Resulting eluate was tested for purity and protein amount on a reducing SDS-PAGE and stained with Imperial protein stain (Thermo Fisher, Cat# 24617). Elutions were then combined and protein concentration was quantified by a Bradford assay (Bio-Rad). Purified protein was then stored at −80°C in a 20% glycerol PBS buffer at 1 mg/ml.

## GST-CRT isolation and TMT mass spectrometry sample preparation

Five 10 cm plates were seeded with 3.5 million cells and allowed to grow for 48 hr. Cells were treated with N-butyldeoxynojirimycin hydrochloride (DNJ) (Cayman Chemicals, Cat # 21065) at 500 µM for 1 hr. Prior to lysis, the media was aspirated and cells were washed once with filter sterilized PBS. Cells were lysed in 1 ml of lysis buffer (20 mM MES, 100 mM NaCl, 30 mM Tris pH 7.5, 0.5% Triton X-100) per plate. Samples were shaken at 4°C for 5 min and centrifuged at 20,800 g at 4°C for 5 min. Lysate was pre-cleared with 25 µl bed volume of buffer-equilibrated glutathione beads per 1

ml of lysate under rotation for 1 hr at 25 µl bed volume. Beads were precipitated by centrifugation at 950 g at 4°C for 5 min. Glutathione beads were pre-incubated with either GST-CRT or GST-CRT-Y109A by equilibrating 25 µl bed volume/pull-down glutathione beads with lysis buffer. Beads were incubated with 100 µg of purified GST-CRT/pull-down under gentle rotation at 4°C for 3 hr and then centrifuged at 950 g at 4°C for 5 min and washed twice with lysis buffer. Supernatant was collected and split in half, with one half incubated for 14 hr at 4°C under gentle rotation with glutathione beads pre-incubated with GST-CRT and the other half under the same conditions with GST-CRT-Y109A.

After incubation with GST-CRT beads, samples were washed once in lysis buffer without protease inhibitors and twice in 100 mM triethylammonium bicarbonate (Thermo Fisher Cat# 90114). After the final wash, samples were incubated with 10 µl of 50 mM DTT (Pierce, Cat# A39255) for 1 hr at room temperature under gentle agitation. Samples were treated with 2 µl of 125 mM iodoacetamide (Pierce, Cat# A39271) and incubated for 20 min under gentle agitation, protected from light. Samples were digested with 5 µg of trypsin (Promega, Cat# V5280) at 37°C overnight under agitation. Peptide concentration was quantified using a BCA protein quantification kit (Pierce, Cat# 23227). 10plex or 6plex TMT (Thermo Fisher 0.8 mg) were resuspended in mass spectrometry grade acetonitrile and was added to digested peptide and incubated for 1 hr at room temperature, per manufacturer's instructions. Labeling was quenched by adding hydroxylamine to 0.25% and incubating for 15 min at room temperature. Labeled samples were pooled, treated with 1,000 units of glycerol-free PNGaseF (NEB, Cat# P0705S), and incubated for 2 hr at 37°C. Samples were cleaned using C18 tips (Pierce, Cat# 87784) and eluted in 75% mass spectrometry grade acetonitrile, 0.1% formic acid (TCI Chemicals). Sample peptide concentration was then quantified using a colorimetric assay (Pierce, Cat# 23275).

## Mass spectrometry data acquisition

An aliquot of each sample equivalent to 3 µg was loaded onto a trap column (Acclaim PepMap 100 pre-column, 75 µm × 2 cm, C18, 3 µm, 100 Å, Thermo Scientific) connected to an analytical column (Acclaim PepMap RSLC column C18 2 µm, 100 Å, 50 cm × 75 µm ID, Thermo Scientific) using the autosampler of an Easy nLC 1000 (Thermo Scientific) with solvent A consisting of 0.1% formic acid in water and solvent B, 0.1% formic acid in acetonitrile. The peptide mixture was gradient eluted into an Orbitrap Fusion mass spectrometer (Thermo Scientific) using a 180 min gradient from 5 to 40%B (A: 0.1% formic acid in water, B:0.1% formic acid in acetonitrile) followed by a 20 min column wash with 100% solvent B. The full scan MS was acquired over range 400–1400 m/z with a resolution of 120,000 (@ m/z 200), AGC target of 5e5 charges, and a maximum ion time of 100 ms and 2 s cycle time. Data-dependent MS/MS scans were acquired in the linear ion trap using CID with a normalized collision energy 35%. For quantitation of scans, synchronous precursor selection was used to select 10 most abundant product ions for subsequent MSthree using AGC target 5e4 and fragmentation using HCD with NCE 55% and resolution in the Orbitrap 60,000. Dynamic exclusion of each precursor ion for 30 s was employed. Data were analyzed using Proteome Discoverer 2.4.1 (Thermo Scientific). Raw spectral data are deposited to MassIVE (ftp://massive.ucsd.edu/MSV000086514/).

## Computational determination of the human N-glycoproteome and substrates analyses

The human N-glycoproteome was defined by the total predicted N-glycosylated proteins from the reviewed human proteome from the UniprotKB (accessed 8/10/2020). Both manual and automated curations of the data set were performed to remove mitochondrial proteins as well as proteins smaller than 50 amino acids from the data set. All annotations were derived directly from the UniprotKB information and annotations available for these proteins were analyzed in R. Determination of the pI values were performed by the pI/MW tool on the Expasy database.

## Reglucosylation validation assay

Five 10 cm plates were seeded with 3.5 million cells each and allowed to grow for 48 hr. Cells were treated with DNJ at 500 µM for 14 hr. Prior to lysis, the media was aspirated and cells were washed once with filter sterilized PBS. Cells were lysed in 1 ml of MNT (20 mM MES, 100 mM NaCl, 30 mM Tris pH 7.5, 0.5% Triton X-100) with protease inhibitors (50 µM Calpain inhibitor I, 1 µM pepstatin,

10 µg/ml aprotinin, 10 µg/ml leupeptin, 400 µM PMSF) and 20 mM N-ethyl maleimide, shaken vigorously for 5 min at 4°C, and centrifuged for 5 min at 17,000 g at 4°C. Fifty microliter bed volume of glutathione beads was added to each pull-down and incubated for 1 hr at 4°C under gentle rotation. Beads were then precipitated by centrifugation at 1000 g for 5 min at 4°C. Supernatant was collected with 10% used for WCL and the remainder split evenly between GST-CRT and GST-CRT-Y109A conjugated glutathione beads, which were generated as previously described, and incubated for 16 hr at 4°C under gentle rotation. Beads were precipitated at 1000 g for 5 min at 4°C. Supernatant was aspirated and beads were washed twice with lysis buffer without protease inhibitors. Beads were treated with reducing sample buffer (30 mM Tris-HCl pH 6.8, 9% SDS, 15% glycerol, 0.05% bromophenol blue). WCLs were trichloroacetic acid (TCA) precipitated by adding TCA to cell lysate to a final concentration of 10%. Cell lysate was then briefly rotated and allowed to incubate on ice for 15 min before centrifugation at 17,000 g for 10 min at 4°C. Supernatants were aspirated and washed twice with cold acetone and centrifuged at 17,000 g for 10 min at 4°C. Supernatants were aspirated and the remaining precipitant was allowed to dry for 5 min at room temperature and briefly at 65°C. Precipitated protein was resuspended in sample buffer. Samples were resolved on a 9% reducing SDS-PAGE and imaged by immunoblotting.

Quantification of immunoblots was conducted using ImageJ software. The amount of protein found in the GST-CRT-Y109A lane was subtracted from the amount of protein in the associated WT GST-CRT lane. This value was then divided by the amount of protein found in the WCL multiplied by 5 to account for the dilution factor and then multiplied by 100. The resulting value yielded the percent reglucosylation in each cell type.

## Metabolic labeling and IGF-1R immunoprecipitation

Two million cells were plated in 6 cm plates and allowed to grow for 40 hr. Cells were pulse labeled for 1 hr with 120 µCi of EasyTag Express$^{35}$S Protein Labeling Mix [$^{35}$S]-Cys/Met (PerkinElmer; Waltham, MA). Immediately after the radioactive pulse, cells were washed with PBS and either lysed in MNT with a protease inhibitor cocktail (Halt protease and phosphatase inhibitor single-use cocktail, Thermo Fisher) and 20 mM NEM, or chased for indicated time using regular growth media. Where indicated, cells were treated with 500 µM DNJ for 30 min prior to [$^{35}$S]-Cys/Met labeling and through the chase. Cell lysates were shaken for 5 min at 4°C, centrifuged at 17,000 g for 5 min at 4°C, and the supernatants were collected. Samples were pre-cleared with a 20 µl bed volume of protein-A sepharose beads (GE Healthcare) by end-over-end rotation for 1 hr at 4°C. The supernatants were collected and incubated with a 30 µl bed volume of protein-A-sepharose beads and 1.5 µl of α-IGF-1 receptor β (D23H3) XP (Cell Signaling) per sample. Samples were washed with MNT without protease inhibitors or NEM and eluted in sample buffer. Samples were then resolved on a 9% reducing SDS-PAGE, imaged using a GE Typhoon FLA 9500 phosphorimager (GE Healthcare), and quantified using ImageJ.

## Glycosylation assay

Three million cells for each indicated cell line were plated in a 10 cm plate and allowed to grow for 48 hr. Cells were lysed in 300 µl RIPA buffer (1% SDS, 1% NP-40, 0.5% sodium deoxycholate, 150 mM NaCl, 50 mM Tris-HCl pH 8.0) with protease inhibitor cocktail and 20 mM NEM. Samples were then sonicated for 20 s at 40% amplitude (Sonics vibra cell VC130PB), shaken vigorously for 5 min, and centrifuged for 5 min at 17,000 g. Twenty microliters of the resulting lysate was heated at 95°C for 5 min and treated with either 10 µl of PNGaseF or EndoH for 1 hr at 37°C, according to the manufacturer's instructions (NEB). Samples were diluted 1:1 into sample buffer and imaged by immunoblotting.

## RNAseq library preparation and sequencing

Three million cells for each indicated cell line were plated in 10 cm plates and allowed to grow for 48 hr. Cells were then lysed in TRIzol buffer and RNA was isolated using RNA Clean Concentrate Kit with in-column DNase-I treatment (Zymo Research Corp), following manufacturer's instructions. The quantity of RNA was assayed on Qubit using RNA BR assay (Life Technologies Corp), and quality was assessed on Agilent 2100 Bioanalyzer using RNA 6000 Nano Assay (Agilent Technologies Inc). Total RNA was used to isolate poly(A) mRNA using NEBNext Poly(A) mRNA Magnetic Isolation

Module, and libraries were prepared using NEBNext UltraII Directional RNA Library Prep Kit for Illumina (New England Biolabs) following manufacturer's instructions. The quantity of library was assayed using Qubit DNA HS assay (Life Technologies Corp), and quality was analyzed on Bioanalyzer (Agilent Technologies Inc). Libraries were sequenced on Illumina NextSeq 500 platform using NextSeq 500/550 High Output v2 kit (150 cycles) with 76 bp paired-end sequencing chemistry.

Sequence quality was assessed using FastQC (*Andrews, 2010*) and MultiQC (*Ewels et al., 2016*). Reads were aligned to the hg38 human reference genome using STAR (*Dobin et al., 2013*). Transcript abundance was quantified using RSEM (*Li and Dewey, 2011*) and normalized to counts per million (CPM) in R using the edgeR software package (*Robinson et al., 2010*). Analyses to compare gene expression between cell types was conducted in Excel by finding the average CPM in the pool of genes of interest for the associated cell type and determining the standard deviation away from the average for each gene of interest. Raw RNAseq data are deposited to NCBI Gene Expression Omnibus at https://www.ncbi.nlm.nih.gov/geo/query/acc.cgi?acc=GSE162262.

## Acknowledgements

This work was supported by the National Institutes of Health under award number (GM086874 to DNH); and a Chemistry-Biology Interface program training grant (T32GM008515 to BMA and NPC). Mass spectral data were obtained at the University of Massachusetts Mass Spectrometry Center (Director Dr. Steve Eyles). Flow cytometry and RNAseq data were conducted at the University of Massachusetts Flow Cytometry Core Facility (Director Dr. Amy Burnside) and the Genomics Resource Laboratory (Director Dr. Ravi Ranjan), respectively. John Swenson conducted the analysis of raw RNAseq data.

## Additional information

### Funding

| Funder | Grant reference number | Author |
| --- | --- | --- |
| National Institute of General Medical Sciences | GM086874 | Daniel N Hebert |
| National Institute of General Medical Sciences | T32GM008515 | Benjamin M Adams<br>Nathan P Canniff |

The funders had no role in study design, data collection and interpretation, or the decision to submit the work for publication.

### Author contributions

Benjamin M Adams, Conceptualization, Resources, Data curation, Formal analysis, Validation, Investigation, Methodology, Writing - original draft, Writing - review and editing; Nathan P Canniff, Data curation, Formal analysis, Validation, Visualization, Writing - review and editing; Kevin P Guay, Resources, Validation, Investigation; Ida Signe Bohse Larsen, Resources; Daniel N Hebert, Conceptualization, Supervision, Funding acquisition, Writing - original draft, Project administration, Writing - review and editing

### Author ORCIDs

Benjamin M Adams https://orcid.org/0000-0002-9980-5321
Daniel N Hebert https://orcid.org/0000-0003-1537-4446

### Decision letter and Author response

Decision letter https://doi.org/10.7554/eLife.63997.sa1
Author response https://doi.org/10.7554/eLife.63997.sa2

## Additional files

### Supplementary files
- Source data 1. Number of glycans per 100 amino acids for UGGT substrates and N-glycoproteome.
- Source data 2. Protein feature analysis of UGGT substrates and N-glycoproteome.
- Supplementary file 1. UGGT1 and UGGT2 expression.
- Supplementary file 2. mRNA expression analysis of UGGT1 and UGGT2 substrates.
- Supplementary file 3. Beta-hexosminidase subunit beta expression trafficking and hypoglycosylation and CI-M6PR hypoglycosylation.
- Supplementary file 4. mRNA expression of lysosomal preferential UGGT2 substrates.
- Transparent reporting form

### Data availability
All data generated during this study are included in the manuscript or supporting files.

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
