## [Decision Letter]

Thank you for submitting your article "Quantitative glycoproteomics reveals substrate selectivity of the ER protein quality control sensors UGGT1 and UGGT2" for consideration by *eLife*. Your article has been reviewed by three peer reviewers, including Elizabeth A Miller as the Reviewing Editor and Reviewer #1, and the evaluation has been overseen by David Ron as the Senior Editor. The following individuals involved in review of your submission have agreed to reveal their identity: John C Christianson (Reviewer #2); Roberto Sitia (Reviewer #3).

The reviewers have discussed the reviews with one another and the Reviewing Editor has drafted this decision to help you prepare a revised submission.

Summary:

All reviewers were enthusiastic about the topic and quality of the work, which will make a substantive contribution to the chaperone and quality control community. However, reviewers had some good suggestions that would broaden the impact of the findings to the more general secretion/trafficking community.

Reviewers 1 and 2 both felt that the analysis of UGGT substrates could be deepened and the representation of the data improved. This would improve the utility of the dataset in terms of understanding of potential mechanisms of engagement and prediction of additional substrates. Specifically, we request further analysis of potential local determinants as suggested by reviewer 2 – domain structure, predicted disorder, etc.

Reviewers 1 and 3 had concerns about the pulse-chase experiments that should be addressed.

Finally, reviewer 3 raises a good point that the folding trajectory of clients might be altered by the absence of ALG6, which would impact recovery as UGGT substrates. Toning down the quantitative nature of the conclusions is probably warranted.

Reviewer #1:

UGGT has largely been studied using model substrates and non-physiological conditions. Here the Hebert lab uses a quantitative proteomics approach to define the full spectrum of endogenous substrates. On the whole the data seem very solid, the topic is important, and this seems like it will be a good resource for the community, but I think the overall impact is could be improved by addressing 2 specific concerns:

1) Figure 4: I question whether these density plots are a good way to present the data. Why not simple scatter plots that show the aa length for the different substrates. Related to this, it's not clear how "significance" was determined. In 4B, the number of glycans might be artifactually high in the UGGT substrate pool because of the nature of the GST purification. The authors claim that since substrates with few glycans were also detected, this isn't a concern, but I think sample bias can't be so simply ignored. Related to this, aa length of substrates may be similarly related to no. of glycans. Correcting for this might be possible.

2) Figure 5: The pulse-chase of IGF-R is essential to show that substrate fate is impacted by UGGT action. But the pulse-chase experiments shown are somewhat difficult for me to interpret. In the UGGT ^-/-^ conditions I don't see much mature form at all, certainly not increasing over time as the precursor would mature. Instead, I see decrease of the precursor, which might indicate degradation? This might amplify the impact and should be considered.

Reviewer #2:

This work by Adams and colleagues describes the identification of native client repertoires of the ER chaperones UGGT_1/2_ family of glucosyltransferases, to better understand the key roles played during glycoprotein biogenesis. The work is well-conceived and executed, while being conveyed in a clear and concise manner. Bioinformatic analysis of identified clients leads the authors to suggest UGGT1 and 2 may prefer different clients with different localisations, topologies and structures. Data on some identified examples (e.g. IGF-1R, ENPP, HEX B) dissects the steps in maturation and role played by UGGT_1/2_ to provide some mechanistic insight but would benefit from a bit more detail. However, these new data do open up the possibilities to better understand the scope of selective responsibilities of reglucosylation by UGGT1 and UGGT2 to govern the maturation efficiency within the glycoproteome.

The authors' clever scheme to isolate UGGT_1/2_ clients using a combination of CRISPR-edited cell lines and lectin-based affinity purification together with quantitative proteomics appears quite powerful and allows them to isolate and identify selectively dependent client proteins, an obviously valuable dataset. A shortcoming might be that the features determined as preferential for UGGT_1/2_ focus on the whole protein are not particularly specific, which leaves the reader wanting a bit more in depth analysis to draw out some potential "local" determinants. While analyses of the clients using UniProt defined features is certainly valid, it means the analysis and predictions are limited and not as detailed as they could have been.

1) It is not clear to this reviewer whether, in the example candidates studded with multiple glycosylation sites, whether it is always a single (or the same) glycan that determines engagement by UGGT1 or 2, or whether it varies and is rather, dependent upon the folded state of the protein. In lieu of performing detailed glycan analysis on clients, perhaps this could be discussed.

2) Moreover, are all glycosylation sites utilised in these clients or only some and does that influence UGGT_1/2_ engagement? Perhaps the authors might address this as an aspect that might help understand selective recognition by UGGT1 or 2.

3) In regard to the UGGT_1/2_ clients identified, are there intrinsic or local folding/maturation features that makes them more frequently in need of reglucosylation than the rest of the glycoproteome? If so, what might that feature be, if not something general like a TMD. Perhaps the authors could further assess the domain structures of clients, or the relative position of glycans within them to add an additional dimension. As a reader, I would like to better understand why these proteins and not the other 97% of glycoproteins enter this route of maturation.

4) Could UGGT activity play a determinant role in multimer assembly, say for the composition of the hexosaminidase dimer, for example where UGGT2 KO cells that reduce efficient trafficking of the HEX B subunit but not HEX A? Does this bias the composition and consequently function? More generally, could the activities of UGGT_1/2_ offer a point of modulation for multimer composition? The authors raised the point of the impact of the UPR in the Discussion, which might be relevant.

5) The authors report that 70% of UGGT1 clients are Type I membrane proteins, but relative to the total number of Type I proteins in the glycoproteome, this number is relatively small. Why these proteins and not the remaining Type I's? Are there unique structural features, folding trajectories or glycan positions that provide some clue as to why these are preferentially engaging UGGT1? (slight reiteration of point 3)

6) If the 3-fold change cut-off is progressively lowered (or raised), how long do the UGGT_1/2_ "preferences" outlined still hold true?

Reviewer #3:

In this clearly written manuscript, Adams et al. set up an ingenious system to identify the clients of UGGT1 and UGGT2. The former is known to act as a folding sensor in the ER lumen adding a glucose moiety to non-native glycoproteins so as to reinsert them in the calnexin/calreticulin cycle. It was not known whether UGGT2 has a role in living cells, and how this would differ from UGGT1.

Key to success was the use of CRISPR to generate cells lacking ALG6, an enzyme in the pathway that generates the oligosaccharide precursors to be transferred to certain asparagines in nascent glycoproteins. In the presence of glucosidase inhibitors, ALG6KO cells accumulated mono-glucosylated proteins only if UGGT1 and/or UGGT2 were present.

An elegant -omic comparison of mono-glucosylated proteins in WT, UGGT1KO, UGGT2KO and double knock outs, allowed the authors to demonstrate that i) both enzymes have activity in vivo; ii) they share some clients; ii) UGGT2 has preferential activity towards small, soluble proteins destined to endo-lysosomes, iii) UGGT1 prefers instead larger plasma-membrane proteins.

As a quantitative and qualitative characterization of the UGGT1-2 clientele is of general interest, the data deserve publication.

Altogether, the data support the conclusions taken. In this reviewer's opinion, however, there is a conceptual problem that the authors should consider and discuss. In the absence of ALG6, glycoprotein substrates are not able to bind calnexin and calreticulin before being glucosylated by the preferred UGGT. As this might shift the folding pathway, many potential clients of UGGT1 or 2 could go undetected. So, in all likelihood the proteins identified are indeed clients of either enzyme, but the quantitative conclusions should be softened and adequately discussed.

Figure 3. Somehow surprisingly, immunoblotting of the whole cell lysates reveals no significant differences in the mature/pro-form ratios in any of the three clients analyzed. This is hard to reconcile with the pulse-chase experiment shown for IGF-1R.The authors may wish to comment about this discrepancy.

Despite sustaining the conclusions taken by the authors, the gels shown in Figures 3I, 3M and 5E are of rather low quality. An effort to improve the aesthetics of the experiments is worth. In Figure 5, a one-hour pulse is quite long to follow the folding of a glycoprotein. A shorter pulse might reveal more details.

[Editors' note: further revisions were suggested prior to acceptance, as described below.]

Thank you for submitting your article "Quantitative glycoproteomics reveals substrate selectivity of the ER protein quality control sensors UGGT1 and UGGT2" for consideration by *eLife*. Your article has been reviewed by three peer reviewers, including Elizabeth A Miller as the Reviewing Editor and Reviewer #1, and the evaluation has been overseen by David Ron as the Senior Editor. The following individuals involved in review of your submission have agreed to reveal their identity: John C Christianson (Reviewer #2); Roberto Sitia (Reviewer #3).

The reviewers have discussed the reviews with one another and the Reviewing Editor has drafted this decision to help you prepare a revised submission.

Summary:

Secretory and membrane proteins are subject to strict quality control, driven in part by engagement with the lectin/chaperone system of the endoplasmic reticulum. Here, the authors have devised an elegant strategy to systematically identify clients that engage either of two separate glycosyltransferases that regulate engagement with this pathway. This analysis provides insight into properties that govern quality control and provide a framework for understanding how protein folding influences secretion.

Revisions:

With apologize for not catching the specific points mentioned by reviewer 1 below, we ask for textual changes that address these concerns. To be clear, we are not asking for additional experiments to be performed, although you may have data from the ALG6/UGGT_1/2_^-/-^ cells already, which would speak to point no. 1. If not, then perhaps you could include an acknowledgement that alternative pathways may contribute, or an explanation of why that is unlikely to be the case.

Reviewer #1:

The revised version is improved and addresses my previous concerns. On re-reading, however, I was struck by a couple of things that might be addressed by the authors textually. Alternatively, I may have missed something…

First, it seems that it would be a good idea to repeat the mass spectrometry in an ALG6/UGGT_1/2_^-/-^ triple KO/KD condition to know that hits recovered in the UGGT single mutants are not non-specific or arising from some redundant enzyme. This is shown in Figure 3 for specific substrates, so it may not be an issue, but it seemed to me to be a potentially important control.

Second, I seem to be missing something with regard to the hits recovered from the ALG6 KO cells versus those with the UGGT enzymes also KO'ed. I would have thought that the ALG6 proteome should encompass all UGGT hits, with smaller numbers of proteins recovered from the single mutants (and none recovered from a double). Yet, there are fewer proteins in the ALG6 ^-/-^ calnexin-precipitated proteome. What am I missing? Is this important?

Finally, in the analysis presented in Figure 4 (which is much easier to interpret now) I wonder if it's worth separating out the lysosomal N-glycoproteome given that the authors claim UGGT clients are more likely to be lysosomal proteins. If one just considers the lysosomal cohort of N-glycome, does this profile more closely resemble the UGGT proteome?

None of these are essential points, but might strengthen an already interesting study.

Reviewer #2:

The revised manuscript has taken on board the comments and suggestions of this reviewer to a satisfactory degree. While it would be desirable for the authors to have been able to say more about the determinants for client selectivity by UGGT_1/2_, this is a complex question and arguably beyond the scope of the current work. The quality of the images and graphs has been improved to better represent the data presented. Moreover, the authors have now included additional statements and/or paragraphs to clarify their results which were unclear or ambiguous. At present, the manuscript will be a valuable resource for the scientific community.

Reviewer #3:

In this reviewer's opinion, the authors provided satisfactory answers to the criticisms raised to the original version, and the data sustain the conclusions reached.

One can always improve the aesthetics of a paper, but there is a moment in which the information package can be considered sufficiently complete. This seems to be the case for this study.

---

## [Author Response]

Reviewers 1 and 2 both felt that the analysis of UGGT substrates could be deepened and the representation of the data improved. This would improve the utility of the dataset in terms of understanding of potential mechanisms of engagement and prediction of additional substrates. Specifically, we request further analysis of potential local determinants as suggested by reviewer 2 – domain structure, predicted disorder, etc.Reviewers 1 and 3 had concerns about the pulse-chase experiments that should be addressed.Finally, reviewer 3 raises a good point that the folding trajectory of clients might be altered by the absence of ALG6, which would impact recovery as UGGT substrates. Toning down the quantitative nature of the conclusions is probably warranted.

We have thoroughly addressed all these points individually below. In short, provided below is: (1) a more in-depth analysis of the properties of the UGGT substrates using a variety of algorithms; (2) results presented as scatter plots rather than density plots; (3) new autorad images for the pulse/chase experiment and immunoblots along with new text that explains the results; and (4) added text that emphasizes the effect that the ALG6 deletion may have on early protein folding steps.

Reviewer #1:UGGT has largely been studied using model substrates and non-physiological conditions. Here the Hebert lab uses a quantitative proteomics approach to define the full spectrum of endogenous substrates. On the whole the data seem very solid, the topic is important, and this seems like it will be a good resource for the community, but I think the overall impact is could be improved by addressing 2 specific concerns:1) Figure 4: I question whether these density plots are a good way to present the data. Why not simple scatter plots that show the aa length for the different substrates. Related to this, it's not clear how "significance" was determined.

As suggested, the data presented in Figure 4 has now been changed to scatter plots with box and whiskers. We agree that this representation provides a more complete display of the results. Thank you. Quartiles and medians are shown.

In 4B, the number of glycans might be artifactually high in the UGGT substrate pool because of the nature of the GST purification. The authors claim that since substrates with few glycans were also detected, this isn't a concern, but I think sample bias can't be so simply ignored.

This is an important point. We have added the following to the text to emphasize this concern: “Finally, the monoglucosylated protein isolation procedure that relies on CST-CRT pulldowns is likely influenced by the number of monoglucosylated sites on a protein. While multiple UGGT substrates with two N-glycans were identified, suggesting heavy glycosylation is not a requirement, the number of monoglucosylated glycans on a substrate is expected to have an impact on the efficiency of substrate isolation and identification.”

Related to this, aa length of substrates may be similarly related to no. of glycans. Correcting for this might be possible.

The number of glycans per 100 amino acids is higher in the UGGT1 and UGGT2 substrate sets as compared to the N-glycoproteome. This suggests that a preference for highly glycosylated protein is not due to the length of the substrates selecting for proteins with a large number of N-glycans, but rather a preference of UGGT1 and UGGT2. Please see the scatter plot in Author response image 1.

2) Figure 5: The pulse-chase of IGF-R is essential to show that substrate fate is impacted by UGGT action. But the pulse-chase experiments shown are somewhat difficult for me to interpret. In the UGGT ^-/-^ conditions I don't see much mature form at all, certainly not increasing over time as the precursor would mature. Instead, I see decrease of the precursor, which might indicate degradation? This might amplify the impact and should be considered.

The quantification of the autorads involves three biological replicates. We have chosen a different autorad image for Figure 5E that better represents the results. Degradation may slightly affect the quantification of the percent of mature IGF-1R, but a consistent issue of degradation was not observed.

Reviewer #2:This work by Adams and colleagues describes the identification of native client repertoires of the ER chaperones UGGT_1/2_ family of glucosyltransferases, to better understand the key roles played during glycoprotein biogenesis. The work is well-conceived and executed, while being conveyed in a clear and concise manner. Bioinformatic analysis of identified clients leads the authors to suggest UGGT1 and 2 may prefer different clients with different localisations, topologies and structures. Data on some identified examples (e.g. IGF-1R, ENPP, HEX B) dissects the steps in maturation and role played by UGGT_1/2_ to provide some mechanistic insight but would benefit from a bit more detail. However, these new data do open up the possibilities to better understand the scope of selective responsibilities of reglucosylation by UGGT1 and UGGT2 to govern the maturation efficiency within the glycoproteome.The authors' clever scheme to isolate UGGT_1/2_ clients using a combination of CRISPR-edited cell lines and lectin-based affinity purification together with quantitative proteomics appears quite powerful and allows them to isolate and identify selectively dependent client proteins, an obviously valuable dataset. A shortcoming might be that the features determined as preferential for UGGT_1/2_ focus on the whole protein are not particularly specific, which leaves the reader wanting a bit more in depth analysis to draw out some potential "local" determinants. While analyses of the clients using UniProt defined features is certainly valid, it means the analysis and predictions are limited and not as detailed as they could have been.

We have used a number of different available algorithms to analyze the UGGT substrates including Pfam (domain numbers), β sheet and α helix propensity (Deleage and Roux), and hydrophobicity (Kyte Doolittle). We had hoped to identify general features of the UGGT substrates that makes then better substrates for the UGGTs; however, there was no obvious properties identified. In future studies, we hope to define specific reglucosylation sites. This will permit us to analyze the local determinants or environment of the modified sites. Plots showing the number of domains predicted by pfam, hydrophobicity, and propensity for β sheet and α helix for each dataset is shown in Author response image 2. The identified UGGT substrates demonstrate a slight increase in predicted domain content, but this is primarily due to a large portion of the N-glycoproteome containing 0 or 1 predicted domain. This likely represents the inability of pfam to accurately predict domains for this large and diverse set of proteins. Pfam also did not predict common domains between UGGT substrates. Little difference between the N-glycoproteome and the substrates could be observed for hydrophobicity, β sheet propensity, or α helix propensity. An excel table detailing the pfam results has been included in Source data 1. For each protein, the type and length of domains are listed. This analysis was not included in the manuscript as no substantive conclusions could be drawn.

**Author response image 2. respfig2:** 

1) It is not clear to this reviewer whether, in the example candidates studded with multiple glycosylation sites, whether it is always a single (or the same) glycan that determines engagement by UGGT1 or 2, or whether it varies and is rather, dependent upon the folded state of the protein. In lieu of performing detailed glycan analysis on clients, perhaps this could be discussed.

This question is of great interest to us. However, our data does not adequately address this question as substrates are affinity purified through GST-CRT pulldown and then trypsinized, resulting in a lack of spatial information. We plan to modify our approach in future studies to attempt to identify the specific sites of reglucosylation. This issue has been discussed in the manuscript through the addition of the following text: “In future studies, we hope to identify specific sites of reglucosylation. These results would demonstrate if reglucosylation occurs on multiple sites or a small number of sites that are influenced by their local environment.”

2) Moreover, are all glycosylation sites utilised in these clients or only some and does that influence UGGT_1/2_ engagement? Perhaps the authors might address this as an aspect that might help understand selective recognition by UGGT1 or 2.

This concern is addressed above in point #1.

3) In regard to the UGGT_1/2_ clients identified, are there intrinsic or local folding/maturation features that makes them more frequently in need of reglucosylation than the rest of the glycoproteome? If so, what might that feature be, if not something general like a TMD. Perhaps the authors could further assess the domain structures of clients, or the relative position of glycans within them to add an additional dimension. As a reader, I would like to better understand why these proteins and not the other 97% of glycoproteins enter this route of maturation.

This concern was addressed above with our description of the algorithms employed to search for defining features of substrates.

4) Could UGGT activity play a determinant role in multimer assembly, say for the composition of the hexosaminidase dimer, for example where UGGT2 KO cells that reduce efficient trafficking of the HEX B subunit but not HEX A? Does this bias the composition and consequently function? More generally, could the activities of UGGT_1/2_ offer a point of modulation for multimer composition? The authors raised the point of the impact of the UPR in the Discussion, which might be relevant.

The following text has been added to address the potential role for UGGT1 and UGGT2 in multimer assembly: “As a majority of the UGGT substrates are found in oligomeric complexes, the UGGTs might also exhibit a preference for unassembled subunits to help ensure proper protein assembly in the ER.”

5) The authors report that 70% of UGGT1 clients are Type I membrane proteins, but relative to the total number of Type I proteins in the glycoproteome, this number is relatively small. Why these proteins and not the remaining Type I's? Are there unique structural features, folding trajectories or glycan positions that provide some clue as to why these are preferentially engaging UGGT1? (slight reiteration of point 3)

To clarify, 70% of the transmembrane (TM) containing UGGT1 hits were type I membrane proteins, not 70% of all UGGT1 hits. About one third of all Type I proteins in the N-glycoproteome pass the characteristics of UGGT1 substrates stated in the manuscript (300+ luminally exposed amino acids, 4+ N-glycans, ≤ 7 pI). All but one of the identified UGGT1 type I proteins possesses these features. A large fraction of type I proteins are poor UGGT1 substrates as they lack the characteristics discussed above. The remaining third of type I proteins may be chaperone-independent folder or UGGT1 substrates not identified by our assay here due to the applied three-fold cutoff, differences in cell-types, the number of reglucosylation sites modified or low expression. In addition, some of these type I membrane proteins may not be UGGT1 substrates due to unknown characteristics. The proteins in the N-glycoproteome that pass this test are provided in Source data 2. This table contains the protein name, accession number and sequence for each N-glycoproteome type I protein that passed the described substrate characteristic test, as well as their values associated with each parameter. These data are not included in the manuscript as a quantitative prediction algorithm requires a larger sample size and further refinement.

6) If the 3-fold change cut-off is progressively lowered (or raised), how long do the UGGT_1/2_ "preferences" outlined still hold true?

Below the applied three-fold cutoff, an increasing percent of proteins that are not predicted to be localized to the secretory pathway are found. As such, the quality of these data are not sufficient to support the determination of the substrate preferences of UGGT_1/2_. We have applied a six-fold cutoff and found that the preferences remain similar to that found for the three-fold cutoff.

Reviewer #3:[…]Altogether, the data support the conclusions taken. In this reviewer's opinion, however, there is a conceptual problem that the authors should consider and discuss. In the absence of ALG6, glycoprotein substrates are not able to bind calnexin and calreticulin before being glucosylated by the preferred UGGT. As this might shift the folding pathway, many potential clients of UGGT1 or 2 could go undetected. So, in all likelihood the proteins identified are indeed clients of either enzyme, but the quantitative conclusions should be softened and adequately discussed.

We agree that this an important issue that we discussed only briefly in the original manuscript but should be emphasized as it greatly impacted our identification of UGGT substrates. We have added the following paragraph to the Discussion: “ALG6^-/-^ cells permitted the trapping of substrates glucosylated by the UGGTs. These cells are also expected to support the enhancement of glucosylation of glycoproteins that are more reliant upon early lectin chaperone intervention. As observed for IGF-1R, the lack of early intervention of the lectin chaperones directed by glucosidase trimming might lead to misfolding; thereby creating a better substrate for the UGGTs. The use of the cell lines lacking the ability to initiate lectin chaperone binding by the glucosidase trimming (Alg6^-/-^) or UGGT reglucosylation (UGGT^-/-^ cells) provides a platform to delineate which part of the lectin chaperone binding cycle has the greatest influence of glycoprotein maturation and trafficking.”

Figure 3. Somehow surprisingly, immunoblotting of the whole cell lysates reveals no significant differences in the mature/pro-form ratios in any of the three clients analyzed. This is hard to reconcile with the pulse-chase experiment shown for IGF-1R.The authors may wish to comment about this discrepancy.

The cell lines used in the immunoblots in Figure 3 are different than those used in the pulse chase experiment in Figure 5. The cell lines in Figure 3 (ALG6/UGGT1^-/-^, ALG6/UGGT2^-/-^, ALG6/UGGT_1/2_^-/-^) were not used in Figure 5. There is likely minimal difference between the cell lines used in Figure 3 as data shown in Figure 5 demonstrates that IGF-1R is most dependent on initial cnx/crt binding, and as such deletion of UGGT1 or UGGT2 in an ALG6^-/-^ background yields minimal changes in IGF-1R folding and trafficking efficiency.

Despite sustaining the conclusions taken by the authors, the gels shown in Figures 3I, 3M and 5E are of rather low quality. An effort to improve the aesthetics of the experiments is worth.

We have now included improved gels for Figures 3I, 3M, and 5E.

In Figure 5, a one-hour pulse is quite long to follow the folding of a glycoprotein. A shorter pulse might reveal more details.

As these experiments are conducted using endogenously expressed proteins with immunoprecipitations using antibodies directed against the endogenous protein rather than tags. A 30-min pulse does not yield a sufficient amount of protein for trafficking analysis. While some details may be lost due to the hour pulse, only a minimal amount of IGF-1R is mature at the 0 hr time point, suggesting an adequately immature population of protein is being examined.

[Editors' note: further revisions were suggested prior to acceptance, as described below.]

Reviewer #1:The revised version is improved and addresses my previous concerns. On re-reading, however, I was struck by a couple of things that might be addressed by the authors textually. Alternatively, I may have missed something.First, it seems that it would be a good idea to repeat the mass spectrometry in an ALG6/UGGT_1/2_^-/-^ triple KO/KD condition to know that hits recovered in the UGGT single mutants are not non-specific or arising from some redundant enzyme. This is shown in Figure 3 for specific substrates, so it may not be an issue, but it seemed to me to be a potentially important control.

To address this concern, we have added the following text: “As reglucosylation was not observed for any of the validated substrates tested when both UGGT1 and UGGT2 were knocked out, these glucosyltransferases appear to be responsible for the reglucosylation of N-glycans in the ER.”

Our reasoning for not performing the TMT labelling mass spec with the triple knockout cell lines is largely one of logistics. This negative control cell line was created a few months after the quantitative mass spec results were obtained with the single and double knockout cell lines. The value of performing the quantitative mass spec as a negative control was greatly diminished when we observed there was no glucosylation of any of the diverse substrates selected for validation in Figure 3. During this period, time in the lab and access to the mass spectrometry became restricted by the pandemic. Therefore, we decided that importance of moving the project and career of the graduate student performing the studies forward significantly outweighed the limited benefit of the additional negative control.

Second, I seem to be missing something with regard to the hits recovered from the ALG6 KO cells versus those with the UGGT enzymes also KO'ed. I would have thought that the ALG6 proteome should encompass all UGGT hits, with smaller numbers of proteins recovered from the single mutants (and none recovered from a double). Yet, there are fewer proteins in the ALG6 ^-/-^ calnexin-precipitated proteome. What am I missing? Is this important?

The explanation for this interesting (and initially surprising) point is found in the text:

“With the *ALG6*/*UGGT2^-/-^* cells, 66 N-glycosylated proteins were identified as reglucosylation substrates using the three-fold cutoff (GST-CRT/CST-CRT-Y109A) (Figure 2A). Nearly double the number of UGGT1 substrates were identified through this approach compared to using *ALG6^-/-^* cells where both UGGT1 and UGGT2 were present. This expansion in substrate number is likely due to the ~50% increase in expression of UGGT1 in *ALG6*/*UGGT2^-/-^* cells (Figure 2—figure supplement 1)…”

Finally, in the analysis presented in Figure 4 (which is much easier to interpret now) I wonder if it's worth separating out the lysosomal N-glycoproteome given that the authors claim UGGT clients are more likely to be lysosomal proteins. If one just considers the lysosomal cohort of N-glycome, does this profile more closely resemble the UGGT proteome?

As suggested, we separated out the lysosome N-glycoproteome to determine if it aligned more closely with the UGGT substrates identified using the three experimental cell lines than the overall N-glycoproteome used in Figure 4. However, as you can see from Author response image 3, the lysosome N-glycoproteome aligns closely with the overall Nglycoproteomes in regard to amino acid length, glycan number and pI. The only parameter where the lysosome Nglycoproteome aligned better with the UGGT substrates was for UGGT2 substrates obtained *from ALG6/UGGT1^-/-^* cells. Therefore, for the manuscript figure, we have chosen to use the complete N-glycoproteome that represents all the N-glycosylated substrates potentially available to the UGGTs for modification.

**Author response image 3. respfig3:**